# A TNFRSF14-FcεRI-mast cell pathway contributes to development of multiple features of asthma pathology in mice

Riccardo Sibilano[1], Nicolas Gaudenzio[1], Marianne K. DeGorter[1,2], Laurent L. Reber[1,3], Joseph D. Hernandez[1,4], Philipp M. Starkl[1,5], Oliwia W. Zurek[1], Mindy Tsai[1], Sonja Zahner[6], Stephen B. Montgomery[1,2], Axel Roers[7], Mitchell Kronenberg[6], Mang Yu[1] & Stephen J. Galli[1,8]

Asthma has multiple features, including airway hyperreactivity, inflammation and remodelling. The TNF superfamily member TNFSF14 (LIGHT), via interactions with the receptor TNFRSF14 (HVEM), can support $T_H2$ cell generation and longevity and promote airway remodelling in mouse models of asthma, but the mechanisms by which TNFSF14 functions in this setting are incompletely understood. Here we find that mouse and human mast cells (MCs) express TNFRSF14 and that TNFSF14:TNFRSF14 interactions can enhance IgE-mediated MC signalling and mediator production. In mouse models of asthma, TNFRSF14 blockade with a neutralizing antibody administered after antigen sensitization, or genetic deletion of *Tnfrsf14*, diminishes plasma levels of antigen-specific $IgG_1$ and IgE antibodies, airway hyperreactivity, airway inflammation and airway remodelling. Finally, by analysing two types of genetically MC-deficient mice after engrafting MCs that either do or do not express TNFRSF14, we show that TNFRSF14 expression on MCs significantly contributes to the development of multiple features of asthma pathology.

[1] Department of Pathology, Stanford University School of Medicine, Stanford, California 94305, USA. [2] Department of Genetics, Stanford University School of Medicine, Stanford, California 94305, USA. [3] Department of Immunology, Unit of Antibodies in Therapy and Pathology, INSERM U1222, Institut Pasteur, Paris 75015, France. [4] Department of Pediatrics, Stanford University School of Medicine, Stanford, California 94305, USA. [5] CeMM Research Center for Molecular Medicine of the Austrian Academy of Sciences and Department of Medicine I, Research Laboratory of Infection Biology, Medical University of Vienna, Vienna 1090, Austria. [6] Division of Developmental Immunology, La Jolla Institute for Allergy and Immunology, La Jolla, California 92037, USA. [7] Institute for Immunology, Technische Universität Dresden, Dresden 01307, Germany. [8] Department of Microbiology and Immunology and Sean N. Parker Center for Allergy and Asthma Research, Stanford University School of Medicine, Stanford, California 94305, USA. Correspondence and requests for materials should be addressed to M.Y. (email: yumyu@stanford.edu) or to S.J.G. (email: sgalli@stanford.edu).

Asthma is a chronic inflammatory disease affecting > 235 million people worldwide and is a major cause of morbidity[1]. Typical features of asthma include shortness of breath, wheezing, airway hyperreactivity (AHR), abundant airway mucus production and structural changes in the airways[2,3]. Severe attacks can cause respiratory insufficiency and death[3]. Both genetic predisposition (atopy) and environmental factors (for example, viruses, allergens, dust and occupational exposures[4]) can contribute to asthma development and progression. In many people asthma has an allergic component, characterized by the $T_H2$ cell-dependent production of antigen (Ag)-specific IgE antibodies that are thought to have a key pathogenic role[5].

In subjects with atopic or 'allergic' asthma, mast cells (MCs) are considered critical for the development of multiple features of the pathology[3,6–9]. This is thought to reflect the MC's ability to release, upon Ag cross-linking of IgE-bound high affinity receptors for IgE (FcεRI), a diverse range of pre-stored and newly synthesized compounds such as histamine, cytokines, chemokines and autacoids, which may, at least in part, initiate or amplify inflammation in situ, increase vascular permeability and contribute to airway remodelling[6,9,10]. Yet the factors which can importantly regulate FcεRI-dependent MC activation in asthma, and thereby influence the nature and magnitude of the MC's roles in the acute and long-term pathology of this disorder, remain to be fully defined[8].

In this context, it has been shown that stimuli from the microenvironment can influence IgE-dependent signalling in MCs and can thereby selectively modulate MC responses, for example, through activation of MC interleukin (IL)-33 receptors (IL-33Rs), thymic stromal lymphopoietin receptors (TSLPRs) and toll-like receptors[11,12]. IgE-dependent signalling in MCs also can be modulated by the engagement of (tumour-necrosis factor (TNF)):TNF receptor (TNFR) superfamily molecules. Notably, TNFRSF9 (4-1BB), CD153, Fas and TNFSF4 (OX40L) have been reported to confer either positive or negative effects on MC effector responses[13–15]. Similarly, several members of the TNF superfamily also can have roles in the development of $T_H2$ responses and/or the pathology of asthmatic airway inflammation, such as TNFSF4 (ref. 16), TNFRSF9 (ref. 17) or TNF itself[18].

Recent data have implicated another member of the TNF superfamily, the ligand TNFSF14 (also known as LIGHT (lymphotoxin-related inducible ligand that competes for glyco-protein D binding to herpesvirus entry mediator on T cells)), in asthma pathology. A study of 242 asthma patients revealed a positive correlation between elevated levels of TNFSF14 in the sputum and impaired lung function (assessed by $FEV_1\%$ predicted[19]). Importantly, TNFSF14 also was identified as a factor which can promote AHR and airway remodelling in mouse models of asthma[20].

TNFSF14 can interact with three receptors in humans, TNFRSF14 (also known as HVEM (the herpes virus entry mediator)); TNFRSF3 (lymphotoxin-beta receptor (LTβR)); and TNFRSF6B (soluble decoy receptor 3 (Dcr3)) and with two receptors in mice (TNFRSF14 and TNFRSF3), suggesting complexity in the potential targets and actions of TNFSF14 in vivo[21]. An analysis of gene expression patterns in nasal lavage specimens from children with asthma found that TNFRSF14 was one of the genes exhibiting higher expression during picorna virus-induced asthma exacerbations compared with values in specimens obtained 7–14 days after the infection[22]. Like other receptors in the TNF superfamily, TNFRSF14 can have pleiotropic functions, including fostering or inhibiting immune responses[23], for example, TNFRSF14:TNFSF14 interactions support the generation and longevity of $T_H2$ cells and promote

$T_H2$ memory through Akt activation[24]. Because $T_H2$ cells can enhance the production of Ag-specific IgE antibodies in response to sensitization with Ag, such effects of TNFSF14 on $T_H2$ cells could contribute to the development of IgE-dependent features of asthma models. However, pharmacological blockade of TNFSF14 with an TNFRSF3-Fc fusion protein diminished allergen-induced airway remodelling in mice even when treatment was initiated after the period of initial Ag sensitization[20], suggesting that additional function(s) of TNFSF14:TNFRSF14 signalling in the complex pathology of asthma may remain to be discovered.

In the present study, we detected TNFRSF14 expression on both human and mouse MCs, and found that TNFSF14-dependent engagement of TNFRSF14 on the MC surface in vitro can potentiate IgE-mediated signalling and can increase significantly the secretion of pre-stored and de novo synthesized MC mediators. We also showed, using both an OVA-induced mouse model of chronic airway inflammation[25] and a house dust mite (HDM)-induced asthma model, and testing two different types of genetically MC-deficient mice, that TNFRSF14 expression specifically on MCs is necessary for the full development of multiple features of asthma pathology in mice, including plasma levels of Ag-specific IgE and $IgG_1$ antibodies, AHR, airway inflammation and airway remodelling. These findings suggest that TNFRSF14 may represent a potential therapeutic target in asthma.

## Results

**TNFSF14 enhances IgE-dependent MC activation via TNFRSF14.** Engagement of other MC membrane co-receptors, such as LFA-1 (ref. 26), CD226 (ref. 27), TNFRSF9 (ref. 13) or TNFSF4 (ref. 15), can either positively or negatively regulate MC activation. It has been reported that bone marrow-derived cultured mouse MCs (BMCMCs) functionally bind TNFSF14 through TNFRSF3, resulting in enhanced production of TNF-α, IL-4, IL-6 and RANTES[28]. However, surprisingly, we detected no expression of TNFRSF3 (or TNFSF14) on MCs from the LAD2 human MC line or on in vitro derived human peripheral blood cultured MCs (huPBCMCs) from $CD34^+$ mononuclear precursors (huPBCMCs), and instead detected strong expression of TNFRSF14 on these two human MC populations (Fig. 1a).

We then questioned whether TNFRSF14-expressing human MCs could respond to TNFSF14 binding to TNFRSF14. IgE-presensitized huPBCMCs displayed enhanced LAMP-1 surface expression (indicative of granule exocytosis[29]) (Fig. 1b) and increased production of pro-inflammatory mediators IL-8 and TNF-α after co-stimulation with anti-IgE and human TNFSF14, but no responses to stimulation with TNFSF14 in the absence of specific Ag (Fig. 1c,d). In vitro stimulation with TNFSF14 in the absence of FcεRI-crosslinking did not detectably influence MC activation, suggesting that TNFRSF14 engagement can contribute to MC activation only in concert with another activation signal, in this case, FcεRI aggregation.

We also performed a single cell analysis of FcεRI and TNFRSF14 activation dynamics in living MCs in real time using time-lapse confocal laser scanning microscopy. We monitored, in three-dimensions (3-D) and at high time resolution, granule secretion by huPBCMCs, as assessed by measuring the fluorescence of the granule-associated marker LAMP-1 (ref. 29), using an AlexaFluor-conjugated anti-human (h)LAMP-1 Ab, LAMP-1-A488 (visualized in green), simultaneously with FcεRI and TNFRSF14 aggregation, using, respectively, AlexaFluor-conjugated anti-IgE (anti-IgE-A650) (visualized in blue) and AlexaFluor-conjugated TNFSF14 (TNFSF14-A594) (visualized in red) (Supplementary Fig. 1a). When TNFSF14-A594 was added to the huPBCMC cultures in the absence of FcεRI aggregation,

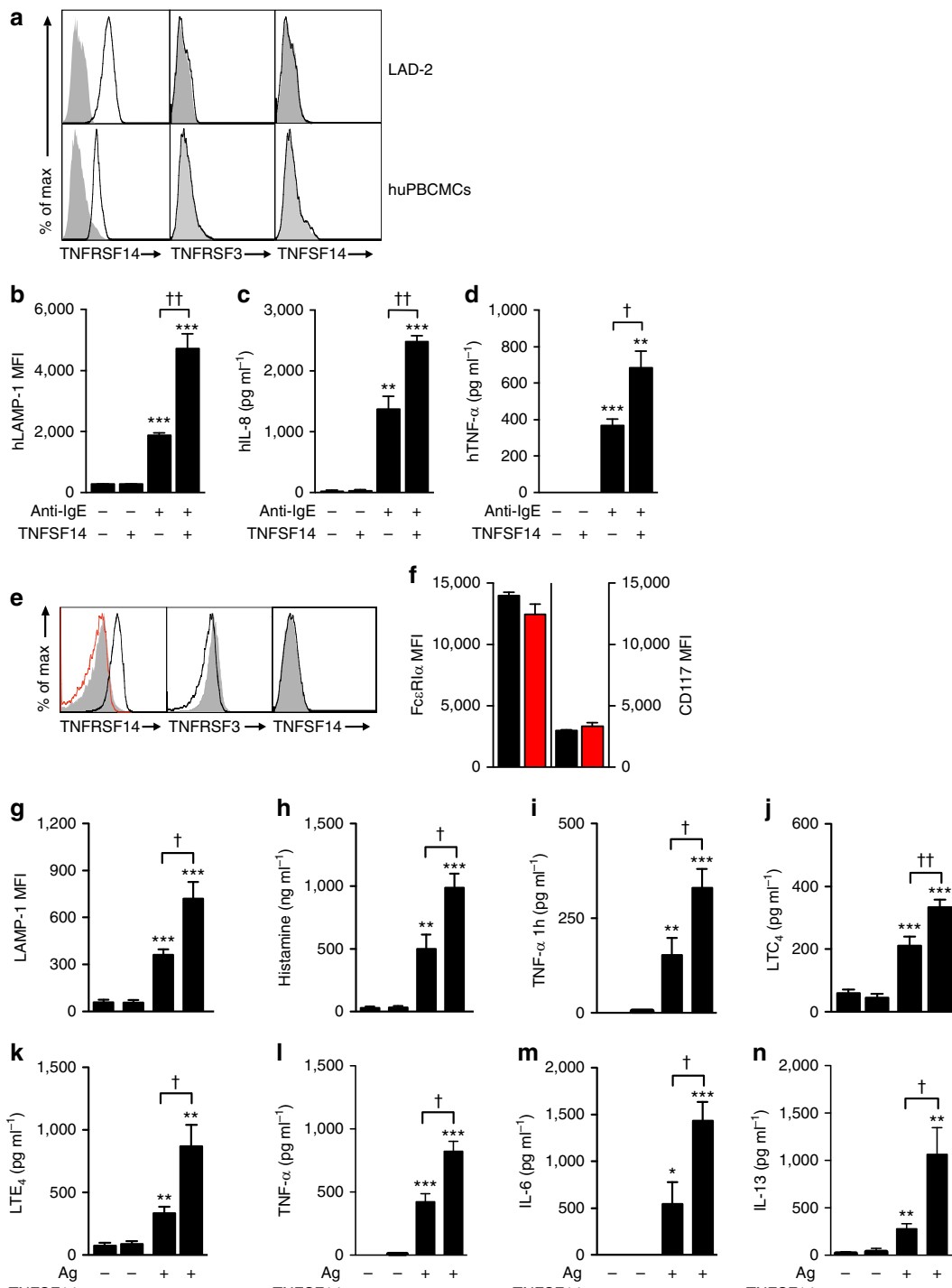

**Figure 1 | TNFRSF14 expression and function on MCs.** (**a**) TNFRSF14, TNFRSF3 and TNFSF14 expression (black lines) on human mast cell line LAD2 cells and human mast cells derived *in vitro* from human peripheral blood CD34$^+$ mononuclear cells (huPBCMCs). Shaded areas: isotype control. (**b–d**) Enhanced IgE-dependent responses upon engagement of huPBCMC TNFRSF14 by TNFSF14. Human (h) LAMP-1 MFI (**b**) and concentrations of hIL-8 (**c**) and hTNF-α (**d**) in the supernatants of IgE presensitized-huPBCMCs, with or without anti-IgE stimulation in the absence or presence of TNFSF14. Results are pooled from three independent experiments, from two donors. (**e**) *Tnfrsf14*$^{+/+}$ mouse bone marrow-derived cultured MCs (BMCMCs), stained for TNFRSF14, TNFRSF3 and TNFSF14 (black lines). Shaded areas: isotype control. Red line indicates TNFRSF14 expression on BMCMCs from *Tnfrsf14*$^{-/-}$ mice. (**f**) FcεRIα$^+$ and CD117$^+$ expression levels (MFI) from three independent cell cultures of *Tnfrsf14*$^{+/+}$ (black columns) and *Tnfrsf14*$^{-/-}$ (red columns) BMCMCs. (**g–n**) LAMP-1 MFI, production of histamine, TNF-α (early 'pre-stored', panel (**i**), later '*de novo* synthesized', panel (**l**)), LTC$_4$, LTE$_4$, IL-6 and IL-13 were measured in the supernatants of IgE presensitized-*Tnfrsf14*$^{+/+}$ BMCMCs, with or without Ag stimulation with or without TNFSF14. Results are pooled from at least four independent experiments, each of which gave similar results. The data in (**b–d,f–n**) (shown as mean + s.e.m.) were assessed for statistical significance using a two-tailed Student's *t*-test. Asterisks indicate statistical significance of differences between non-Ag-treated and corresponding Ag-treated groups; daggers indicate statistical significance between indicated groups. $^*$ or $^†P < 0.05$; $^{**}$ or $^{††}P < 0.01$; $^{***}P < 0.001$.

only a modest number of TNFRSF14/TNFSF14-A594 aggregates were formed on the huPBCMC surface and, consistent with the data in Fig. 1b–d, no LAMP-1 signals were detected (Supplementary Fig. 1a). The addition of anti-IgE-A650 induced formation of some FcεRI/anti-IgE-A650 clusters and rapid generation of LAMP-1 signals (Supplementary Fig. 1a). When anti-IgE and TNFSF14 were added simultaneously, we observed formation of substantially higher numbers of both FcεRI/anti-IgE-A650 and TNFRSF14/TNFSF14-A594 clusters together with enhanced LAMP-1 fluorescence, indicating a synergistic effect of FcεRI-clustering and TNFSF14 on huPBCMC degranulation (Supplementary Fig. 1a).

To analyse precisely the number and dimension of clusters on the surface of individual huPBCMCs, we modelled both FcεRI/anti-IgE-A650 and TNFRSF14/TNFSF14-A594 clusters 30 min after stimulation. Combining anti-IgE-A650 with TNFSF14 dramatically increased both the number and the area of individual clusters of FcεRI/anti-IgE-A650 and TNFRSF14/TNFSF14-A594 on the plasma membrane surface (Supplementary Fig. 1b–e).

These results were confirmed in mouse MCs (mouse bone marrow-derived cultured MCs (BMCMCs)). We detected no expression of TNFRSF3 (or TNFSF14) on BMCMCs generated in vitro from WT $Tnfrsf14^{+/+}$ (that is, TNFRSF14-expressing) mice (Fig. 1e). BMCMCs generated from both $Tnfrsf14^{+/+}$ mice and $Tnfrsf14^{-/-}$ mice had similar levels of expression of FcεRIα and CD117 (KIT, Fig. 1f). We then evaluated whether TNFSF14:TNFRSF14 interactions could influence IgE-dependent MC activation on TNFRSF14-expressing mouse BMCMCs, like we found they did in human MCs. BMCMCs from $Tnfrsf14^{+/+}$ or $Tnfrsf14^{-/-}$ mice were sensitized overnight with an anti-dinitrophenol (DNP) monoclonal mouse IgE antibody and then challenged with DNP-HSA (Ag) in the absence or presence of soluble TNFSF14.

The IgE/Ag-induced production of several stored or newly synthesized mediators was quite similar in $Tnfrsf14^{+/+}$ and $Tnfrsf14^{-/-}$ BMCMCs in the absence of TNFSF14 (Fig. 1g–n (third bars) and Supplementary Fig. 2a–h (third bars)). By contrast, only $Tnfrsf14^{+/+}$ BMCMCs exhibited TNFSF14-dependent enhancement of Ag-mediated activation and mediator production (Fig. 1g–n (fourth bars) and Supplementary Fig. 2a–h (fourth bars)). The lack of responsiveness of $Tnfrsf14^{-/-}$ BMCMCs to TNFSF14 argues against the possibility that BMCMC expression of undetectably low levels of the alternate TNFSF14 receptor, TNFRSF3, can influence IgE/Ag activation of such BMCMCs.

MCs represent a potentially important source of pro-inflammatory and pro-remodelling mediators after Ag challenge in allergic subjects[30,31]. We found that in the presence of FcεRI engagement MC-TNFRSF14:TNFSF14 interactions enhanced: (1) MC surface expression of the granule-associated marker LAMP-1 (ref. 29) (Fig. 1g); (2) MC release of pre-stored mediators (pre-synthesized histamine and TNF-α, Fig. 1h,i); and (3) MC production of several other mediators which could contribute to the development of asthma pathology: $LTC_4$, $LTE_4$, TNF-α, IL-6 and IL-13 (Fig. 1j–n). Notably, BMCMCs stimulated with TNFSF14 in the absence of Ag did not exhibit mediator production significantly different than that seen in control cells stimulated with medium, indicating that TNFRSF14 engagement by TNFSF14 does not activate MC mediator production as a single signal. We considered the possibility that BMCMCs could themselves produce TNFSF14 after Ag or Ag/TNFSF14 stimulation. Although we detected TNFSF14 messenger RNA in $Tnfrsf14^{+/+}$ BMCMCs in all the conditions analysed, including in cells not exposed to Ag or TNFSF14 stimulation (Supplementary Fig. 2i and Supplementary Fig. 3), we were not

able to detect any TNFSF14 protein in the cells' supernatant by enzyme-linked immunosorbent assay (ELISA) or on the cells' surface by flow cytometry (data not shown).

We also modelled both FcεRI/Ag (DNP-HSA-A650, blue) and TNFRSF14/TNFSF14-A594 (red) clusters after stimulation of mouse BMCMCs with TNFSF14, IgE and antigen, or both stimuli. Exposure of IgE-sensitized mouse BMCMCs to both DNP-HSA-A650 and TNFSF14 dramatically increased both the number and area of individual clusters formed by FcεRI and TNFSF14 on the plasma membrane surface (Supplementary Fig. 4a–e). Moreover, we did not observe any binding of TNFSF14 on the surface of $Tnfrsf14^{-/-}$ BMCMCs, indicating that TNFSF14 binding is specific for MCs expressing TNFRSF14 (Supplementary Fig. 5a). Taken together, our results with both human PBCMCs and mouse BMCMCs indicate that engagement of TNFSF14 by TNFRSF14 on the MC surface can enhance the IgE-dependent aggregation of FcεRI.

**TNFSF14 can enhance FcεRI-dependent Akt-PKCδ-MAPK activation**. In vitro stimulation with TNFSF14 in the absence of Ag did not detectably influence MC activation (Fig. 1b–d,g–n), suggesting that TNFRSF14 engagement can contribute to MC activation only in concert with another activation signal, in this case, FcεRI aggregation. MC activation initiated by aggregation of FcεRI requires the recruitment of additional signalling molecules to the MC membrane[29,32]. In Ag-activated $Tnfrsf14^{+/+}$ BMCMCs but not $Tnfrsf14^{-/-}$ BMCMCs (Fig. 2a,b, Supplementary Fig. 5b and Supplementary Figs 6–8), TNFSF14 induced enhanced phosphorylation of specific activating residues of proteins in the Akt-PKCδ-Ras-Mek-Erk pathway, which is importantly involved in MC degranulation and the production of leukotrienes and cytokines, for example, TNF-α, IL-6 and IL-13 (refs 33–35). In accord with the data on BMCMC mediator production shown in Fig. 1g–n (second bar), stimulation of BMCMCs with TNFSF14 in the absence of Ag did not result in phosphorylation of Akt-PKCδ-Ras-Mek-Erk pathway proteins (Fig. 2a).

**Inhibiting TNFRSF14 markedly reduces asthma pathology**. This possible involvement of TNFRSF14 (whether on MCs or other cell types) in features of asthma inflammation and remodelling has not before been specifically investigated. Doherty et al.[20] employed treatment with a TNFRSF3-Fc fusion protein to antagonize actions of TNFSF14 in their asthma model, and this reagent could have interfered with actions of TNFSF14 on either TNFRSF3 or TNFRSF14, as well as block actions of lymphotoxin. Interestingly, treatment with the TNFRSF3-Fc fusion protein was associated with resolution of the features of tissue remodelling observed in this setting, without substantially influencing the extent of leukocyte infiltration in the lung[20].

To investigate the effect, in a mouse asthma model, of specifically targeting TNFRSF14, we first tested a mouse model of OVA-induced chronic airway inflammation (Fig. 3a), which requires for its full development MCs and the γ-chain of the FcεRI (FcεRIγ)[25,36]. Compared with PBS-treated mice, OVA sensitized and challenged C57BL/6J mice exhibited significantly increased: (1) changes in lung resistance ($R_L$) and dynamic compliance ($C_{dyn}$) (Fig. 3b and Supplementary Fig. 9a) upon methacholine (Mch) challenge; (2) numbers of bronchoalveolar lavage (BAL) fluid monocytes, neutrophils, eosinophils, total lymphocytes and $T_H1$, $T_H2$ and $T_H17$ lymphocytes (Fig. 3c and Supplementary Fig. 9b); 3) blood levels of OVA-specific $IgG_1$ and IgE antibodies (Fig. 3d,e); (4) levels of lung collagen (Fig. 3f); (5) numbers of mucus-producing goblet cells in the airway epithelium (Supplementary Fig. 9c); (6) BAL fluid mucin (Muc5AC, Supplementary Fig. 9d)

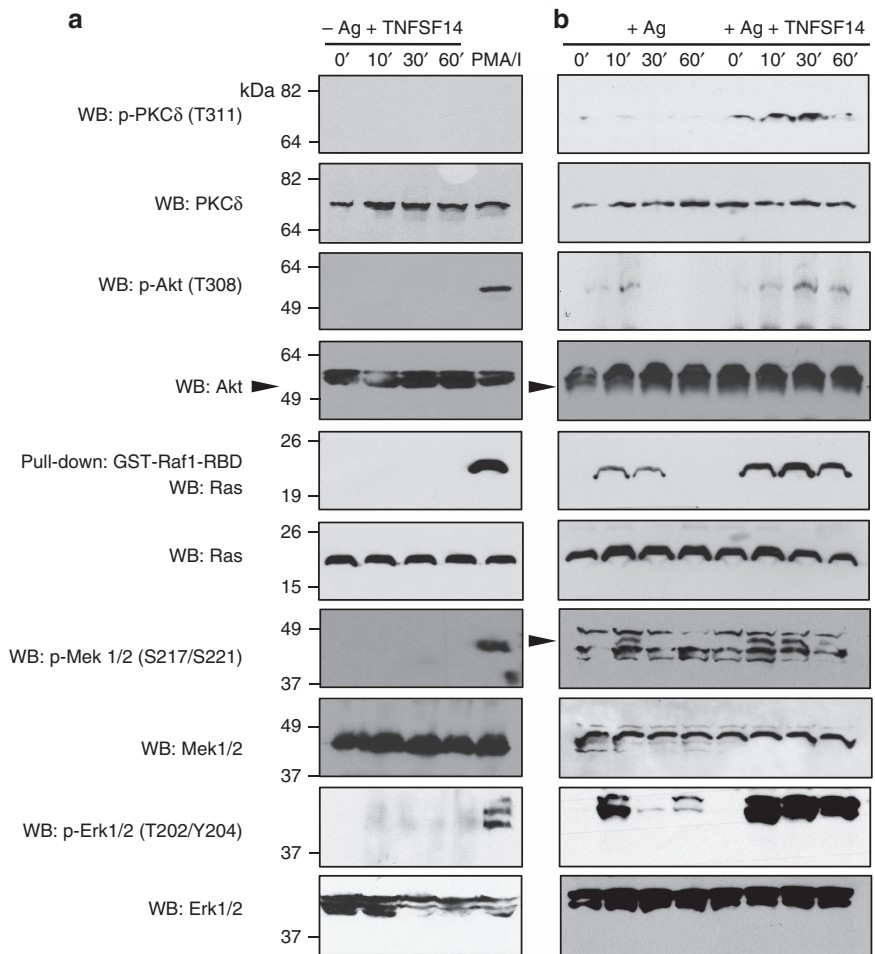

**Figure 2 | TNFSF14 enhances IgE-dependent signalling in MCs.** (**a**) Detection of phosphorylated Akt, PKC-δ, Mek, Erk1/2 and pulled-down Ras, 0, 10, 30, 60 min after stimulation with TNFSF14 in IgE-presensitized $Tnfrsf14^{+/+}$ BMCMCs in the absence of Ag stimulation. Phorbol 12-myristate 13-acetate/ Ionomycin (PMA/I) stimulation (10 min): positive control for mast cell activation. The blots were stripped and reprobed with antibodies recognizing total levels of Akt, PKC-δ, Mek and Erk1/2 for loading control. Loading control for Ras was collected from total BMCMC lysate before glutathione S-transferase (GST)-pulldown. (**b**) Detection of phospho-proteins as in (**a**) 0, 10, 30, 60 min after Ag stimulation or Ag/TNFSF14 co-stimulation of $Tnfrsf14^{+/+}$ BMCMCs. All the blots were stripped and reprobed as in **a**. Loading control for Ras was collected as in **a**. Results shown are representative of three independent experiments, each of which gave similar results.

and TNFSF14 (Supplementary Fig. 9e); and (7) numbers of lung MCs (Fig. 3g). Histological analysis (with representative panels shown in Fig. 3h), demonstrated that this OVA model also was associated with leukocyte infiltrates in the lungs (haematoxylin and eosin staining, upper panel) and detection of airway goblet cells (Masson's Trichrome staining, middle panel), both of which were diminished after treatment with an anti-TNFRSF14 Ab, and with the presence of lung MCs (Toluidine Blue staining, lower panel).

In comparison to the effects of an isotype control antibody, two intraperitoneal injections of an anti-TNFRSF14 blocking antibody (clone LH1, which specifically blocks TNFRSF14:TNFSF14 interactions[37]), given after Ag sensitization at 1 h before the eighth and ninth intranasal OVA challenges, resulted in significant reductions in virtually all of the features that were assessed (Fig. 3 and Supplementary Fig. 9). In addition to reducing several features of airway inflammation and remodelling in this model, anti-TNFRSF14 antibody treatment also was associated with decreased plasma concentrations of OVA-specific IgG$_1$ and IgE antibodies (Fig. 3d,e), indicating that some effect(s) of TNFRSF14 blockade can influence systemic levels of these Ag-specific antibodies, even when the anti-TNFRSF14 antibody was administered in this model long after initial Ag sensitization.

These results were corroborated in a subsequent experiment in which blood was collected 1 h after each intranasal (i.n.) OVA challenge. After the seventh OVA challenge, the initial cohort of mice was split into three groups (no Ab treatment, anti-TNFRSF14 Ab treatment and Iso Ctrl Ab treatment) and at the end of the ninth i.n. OVA challenge, levels of OVA-specific IgG$_1$ and IgE collected during the nine i.n. challenges were quantified. Results shown in Supplementary Fig. 9f–i show that blockade of the TNFRSF14:TNFSF14 axis during the last week of the model lowered the blood concentrations of OVA-specific IgG$_1$ and IgE Abs. These observations might reflect direct actions of the anti-TNFRSF14 Ab blockade on B cells[23], as well as direct and/or indirect effect(s) of such treatment on other TNFRSF14$^+$ cell types, including MCs.

Consistent with our findings in anti-TNFRSF14 antibody-treated WT mice, $Tnfrsf14^{-/-}$ mice that genetically lacked TNFRSF14 in all cell types exhibited decreased: (1) airway responses to Mch ($R_L$ (Fig. 4a) and $C_{dyn}$ (Supplementary Fig. 10a)); (2) OVA-induced increases in BAL leukocytes (Fig. 4b and Supplementary Fig. 10b); (3) blood levels of OVA-specific IgE and IgG$_1$ (Fig. 4c,d); (4) levels of lung collagen (Fig. 4e); (5) numbers of mucus-producing goblet cells in the airway epithelium (Supplementary Fig. 10c); (6) concentrations of

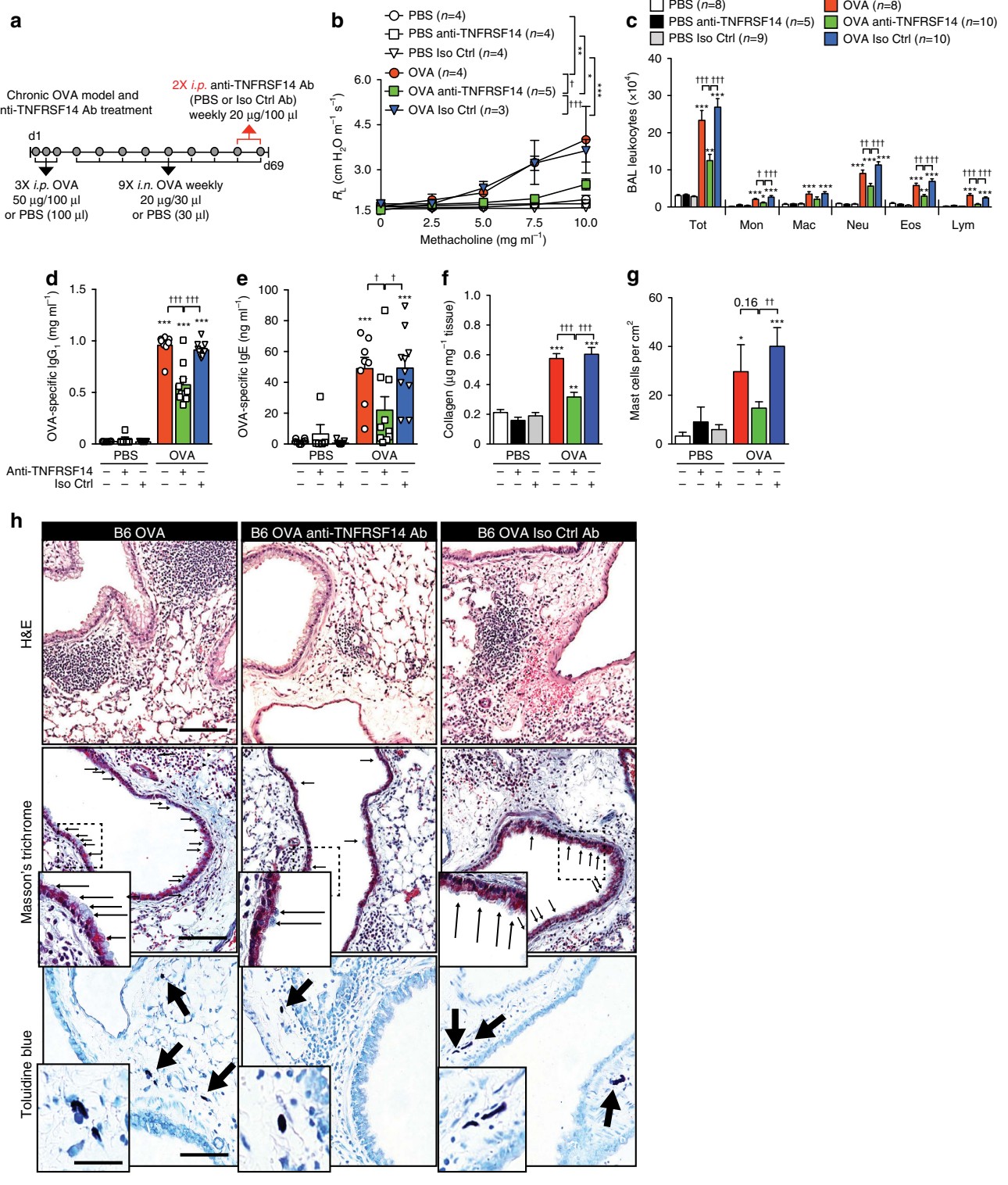

**Figure 3 | Effects of TNFRSF14 blockade after OVA sensitization.** (**a**) Experimental protocol. C57BL/6J mice were sensitized with three intraperitoneal (i.p.) injections of OVA or PBS, followed by 9 weekly intranasal (i.n.) challenges with OVA or PBS. Anti-TNFRSF14 or isotype control (Iso Ctrl) antibody or PBS were injected i.p. One hour before the eighth and ninth i.n. OVA (or PBS) challenges. (**b**) Changes in $R_L$ induced by aerosolized methacholine, (**c**) numbers of leukocytes in BAL fluid (Tot, Total; Mon, monocytes; Mac, macrophages; Neu, neutrophils; Eos, eosinophils; Lym, lymphocytes), (**d,e**) Levels of plasma OVA-specific IgG$_1$ (**d**) and IgE (**e**), (**f**) levels of lung collagen and (**g**) number of MCs, 24 h after the ninth OVA or PBS challenge. Four-to-ten female mice per group were used as indicated. Results are pooled from three independent experiments, each of which gave similar results. (**h**) Representative lung sections stained with haematoxylin and eosin (H&E) (upper panel), Masson's Trichrome (middle panel), demonstrating collagen (blue staining) and goblet cells (small arrows), or Toluidine blue (lower panel), demonstrating MCs (arrows) in mice 24 h after the ninth OVA challenge. Scale bars, 100 μm (insets: 25 μm). The data in **b–g** (shown as mean + s.e.m.) were assessed for statistical significance using a two-tailed Student's t-test or a two-way analysis of variance (ANOVA) test. Asterisks indicate statistical significance of differences between PBS-treated and corresponding Ag-treated groups; daggers indicate statistical significance between indicated groups. * or †P < 0.05; ** or ††P < 0.01; *** or †††P < 0.001.

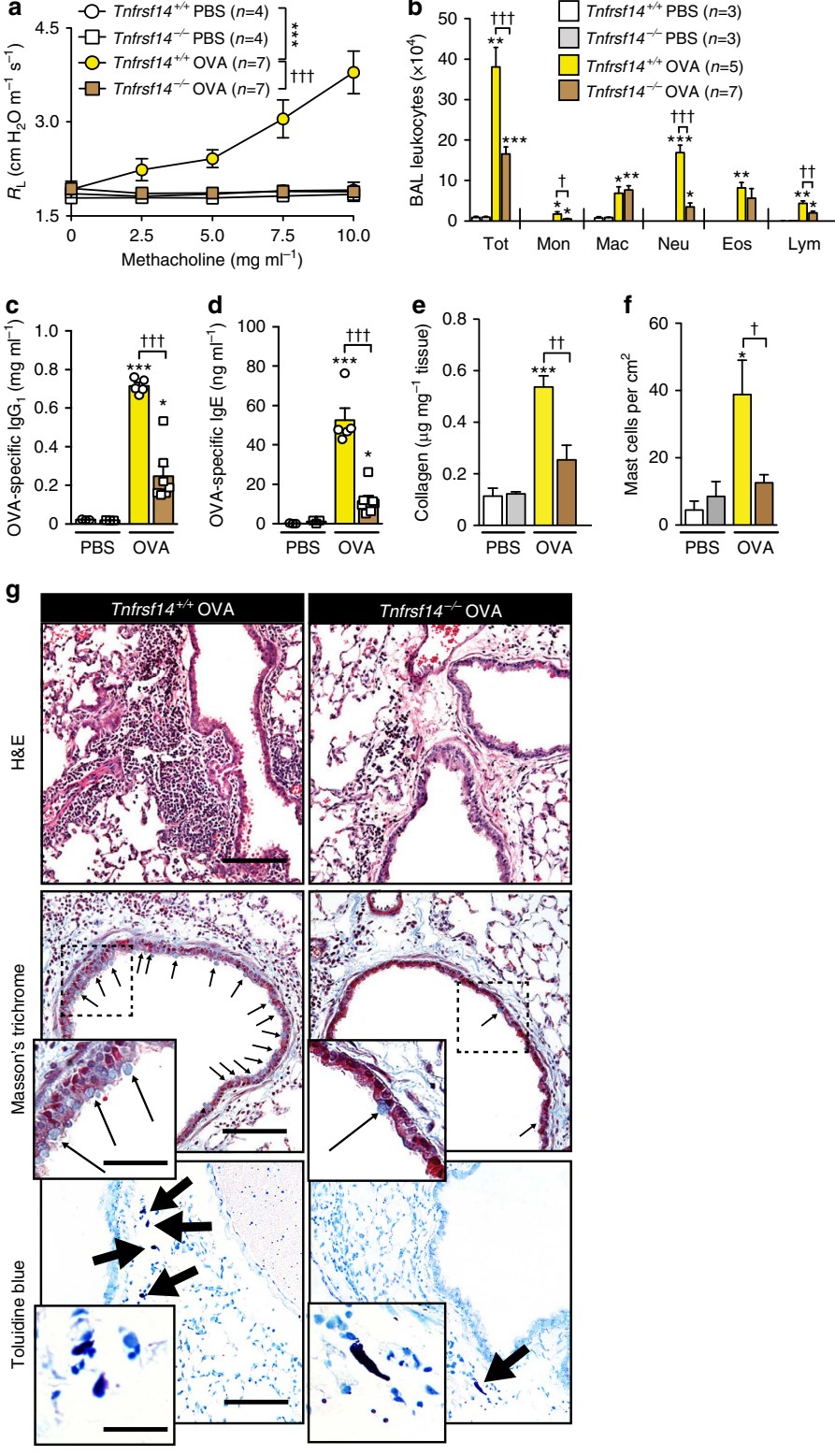

**Figure 4 | $Tnfrsf14^{-/-}$ mice exhibit diminished OVA-induced asthma pathology.** (**a**) Changes in $R_L$ induced by aerosolized methacholine, (**b**) numbers of leukocytes in BAL fluid, (**c,d**) levels of plasma OVA-specific $IgG_1$ (**c**) and IgE (**d**), (**e**) levels of lung collagen and (**f**) numbers of lung mast cells, 24 h after the ninth OVA (or PBS) challenge in $Tnfrsf14^{-/-}$ and littermate control $Tnfrsf14^{+/+}$ mice. Three-to-seven female mice per group were used as indicated. Results are pooled from three independent experiments, each of which gave similar results. (**g**) Representative lung sections stained with haematoxylin and eosin (H&E) (upper panel), Masson's Trichrome (middle panel), demonstrating collagen (blue staining) and goblet cells (small arrows), or Toluidine blue (lower panel), demonstrating MCs (arrows) 24 h after the ninth OVA challenge. Three-to-seven female mice per group were used as indicated. Scale bars, 100 µm (insets: 25 µm). The data in **a**-**f** (shown as mean + or ± s.e.m.) were assessed for statistical significance using a two-tailed Student's $t$-test or a two-way analysis of variance (ANOVA) test. Asterisks indicate statistical significance of differences between PBS-treated and corresponding Ag-treated groups; daggers indicate statistical significance between indicated groups. * or $^{†}P < 0.05$; ** or $^{††}P < 0.01$; *** or $^{†††}P < 0.001$.

BAL Muc5AC and TNFSF14 (Supplementary Fig. 10d,e); and (7) numbers of lung MCs (Fig. 4f). The histological findings are shown in Fig. 4g.

Taken together, our observations in WT mice treated with an anti-TNFSF14 antibody weeks after the induction of Ag sensitization, and in $Tnfrsf14^{-/-}$ mice, indicate that TNFRSF14 is required for the full development of many features of the disease, including blood levels of Ag-specific IgE and IgG$_1$ antibodies, enhanced airway responses to Ag and Mch and features of both airway inflammation and airway remodelling.

**MC TNFRSF14 expression can exacerbate asthma pathology.** To define the importance of MC expression of TNFRSF14 *in vivo*, we induced our model of asthma in genetically MC-deficient $Kit^{W-sh/W-sh}$ mice engrafted intravenously (i.v.) with $Tnfrsf14^{+/+}$ or $Tnfrsf14^{-/-}$ BMCMCs. OVA sensitized and challenged $Kit^{W-sh/W-sh}$ mice that had been engrafted with $Tnfrsf14^{-/-}$ BMCMCs, compared with identically treated $Kit^{W-sh/W-sh}$ mice that had been engrafted with $Tnfrsf14^{+/+}$ (TNFRSF14$^+$) BMCMCs, exhibited significantly reduced: (1) airway responses to Mch (Fig. 5a and Supplementary Fig. 11a); (2) levels of BAL leukocytes after OVA challenge (Fig. 5b and Supplementary Fig. 11b); (3) blood levels of OVA-specific IgG$_1$ and IgE (Fig. 5c,d); (4) levels of lung collagen (Fig. 5e); (5) numbers of mucus-producing goblet cells in the airway epithelium (Supplementary Fig. 11c); and (6) concentrations of BAL Muc5AC and TNFSF14 (Supplementary Fig. 11d,e). Histology confirmed reduced levels of leukocyte infiltration and numbers of goblet cells (Fig. 5g, upper and middle panels). Importantly, the OVA sensitized and challenged $Kit^{W-sh/W-sh}$ mice that had been engrafted i.v. with $Tnfrsf14^{+/+}$ or $Tnfrsf14^{-/-}$ BMCMCs exhibited similar numbers of MCs in the lungs (Fig. 5f and Fig. 5g (lower panel)). As previously reported[36], OVA sensitization and challenge significantly increased numbers of lung MCs in both the groups of BMCMC-engrafted $Kit^{W-sh/W-sh}$ mice compared with values in PBS mock sensitized and challenged mice (Fig. 5f,g (lower panels)), and the MC-engrafted $Kit^{W-sh/W-sh}$ mice exhibited more MCs in the periphery of the lungs than did WT mice.

Similar results were obtained when we engrafted TNFRSF14$^+$ or TNFRSF14-deficient MCs into a different type of mouse which is markedly MC- (and basophil) deficient independently of genetic abnormalities affecting c-kit, namely C57BL/6-$Cpa3$-$Cre$; $Mcl$-$1^{fl/fl}$ mice[38] (Supplementary Fig. 12a–j). Our results, obtained in both 'c-kit-dependent' and 'c-kit-independent' MC-deficient mice thus strongly support the conclusion that even though TNFRSF14 can be expressed on multiple cell types, TNFRSF14 expression on MCs is necessary for the full development of multiple features of asthma pathology in our OVA model.

**Transfer of OVA responses with sera from Tnfrsf14$^{-/-}$ mice.** We considered the possibility that the abnormalities in the pathology of the asthma models which we observed in mice genetically deficient in TNFRSF14 might reflect effects of TNFRSF14 on the sensitization and/or effector phases of T$_H$2 responses. Indeed, we found that OVA sensitized and challenged mice that were globally TNFRSF14-deficient or that contained TNFRSF14-deficient MCs exhibited significantly lower levels of Ag-specific IgE and IgG$_1$ than did the corresponding control mice (Figs 4c,d and 5c,d). Moreover, even WT mice treated with an anti-TNFRSF14 antibody weeks after initial OVA sensitization exhibited significantly lower levels of Ag-specific IgE and IgG$_1$ than did the corresponding isotype antibody-treated control mice (Fig. 3d,e). We, therefore, used a passive sensitization approach to test whether the low levels of Ag-specific IgE present in such mice

were adequate to sensitize effector cells to orchestrate Ag-dependent airway responses.

We found that serum from OVA-sensitized $Tnfrsf14^{-/-}$ mice or from mice containing only $Tnfrsf14^{-/-}$ MCs (Fig. 6a) was as effective as serum from the corresponding WT mice or mice containing TNFRSF14$^+$ MCs in passively sensitizing recipient C57BL/6J mice to exhibit strong airway responses to OVA challenge (Fig. 6b,c). These data show that even the low amounts of OVA-specific IgE produced in mice lacking TNFRSF14, or containing MCs which lack TNFRSF14, are sufficient to mediate IgE- and MC-dependent airway responses. It is possible that some of the effects of the passive transfer of such sera also may reflect activation of MCs by immune complexes of passively transferred OVA-specific IgG$_1$ and OVA[39], but, in general, such effects of IgG$_1$-Ag immune complexes are seen when very large amounts of Ag-specific IgG$_1$ are present[40]. Our results are consistent with the well-accepted notion that it is IgE bound to the surface of effector cell FcεRIs, rather than the additional soluble IgE present in the circulation, that mediates the MC-dependent biological responses to Ag challenge in such settings[41].

We then assessed whether expression of TNFRSF14 solely on MCs is sufficient to enhance the features of this asthma model. To do this, we engrafted $Tnfrsf14^{-/-}$ mice and the littermate control $Tnfrsf14^{+/+}$ mice with $Mcpt5$-$eYFP$ BMCMCs. In $Mcpt5$-$Cre$ transgenic mice, Cre is expressed under the control of the MC protease (Mcpt) 5 promoter. $Mcpt5$-$Cre$ transgenic mice were crossed with Cre-inducible ROSA-$eYFP$ reporter mice, as detailed in the Methods section. BMCMCs derived from such mice are thus TNFRSF14$^+$ (since they express a WT $Tnfrsf14$ gene) and specifically express eYFP. We found that the $Tnfrsf14^{-/-}$ mice in which only the engrafted MCs expressed TNFRSF14 exhibited multiple features of the OVA-induced asthma model, including blood levels of OVA-specific IgG$_1$ and IgE, that were not significantly different than those in the corresponding BMCMC-engrafted $Tnfrsf14^{+/+}$ mice (Fig. 7a–g and Supplementary Fig. 13a–c).

**MC TNFRSF14 exacerbates HDM-induced asthma pathology.** To evaluate whether the data obtained in our OVA model could be confirmed in a second model of asthma, we sensitized $Tnfrsf14^{+/+}$ or $Tnfrsf14^{-/-}$ mice i.n. with HDM from $Dermatophagoides\ pteronyssinus$ and challenged them i.n. weekly for 10 weeks with HDM (Fig. 8a). In this HDM model, compared with $Tnfrsf14^{+/+}$ mice, $Tnfrsf14^{-/-}$ mice exhibited decreased $R_L$ and $C_{dyn}$ responses to Mch (Fig. 8b and Supplementary Fig. 14a), reduced numbers of BAL leukocytes (Fig. 8c), and lower levels of: Ag-specific IgG$_1$ and IgE (Fig. 8d,e), lung collagen (Fig. 8f), BAL Muc5AC and TNFSF14 (Supplementary Fig. 14b,c) and numbers of lung MCs (Fig. 8g). Similarly, experiments performed in MC-deficient $Kit^{W-sh/W-sh}$ mice and $Kit^{W-sh/W-sh}$ mice engrafted with $Tnfrsf14^{+/+}$ or $Tnfrsf14^{-/-}$ BMCMCs showed that MCs and MC expression of TNFRSF14 can contribute significantly to multiple features of this HDM model in these mice (Fig. 8h–m and Supplementary Fig. 14d–f).

**Discussion**
We found that mouse and human MCs express TNFRSF14 (and not TNFRSF3) on their surface and that TNFSF14-mediated engagement of TNFRSF14 enhances activation of such MCs when they are co-stimulated by IgE-dependent mechanisms. *In vivo*, the TNFSF14 receptor TNFRSF14 is required for the full development of many aspects of two mouse models of Ag-induced chronic airway inflammation, an OVA model which recapitulates many of the functional, immunological and gene expression changes that are observed in human 'allergic'

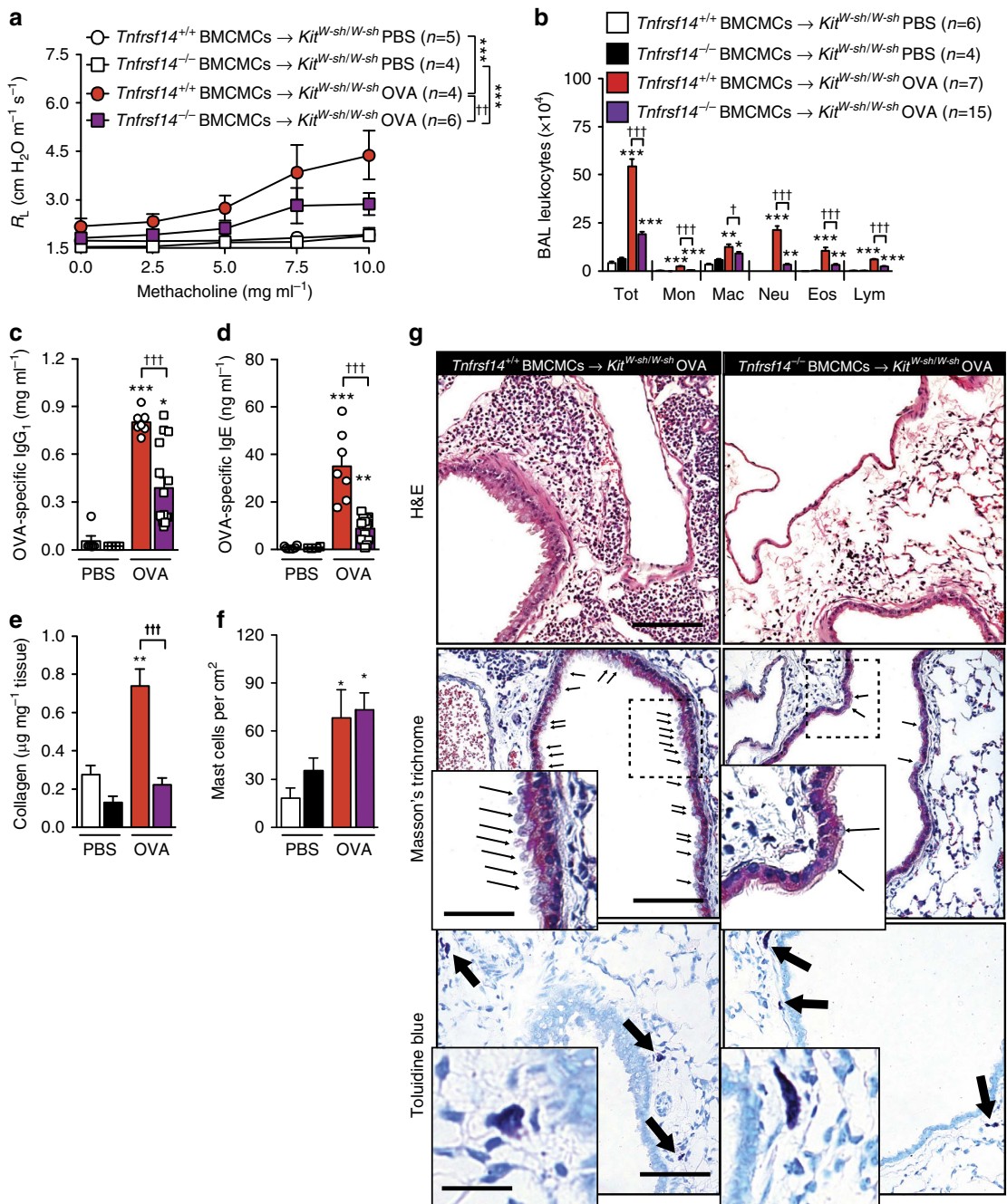

**Figure 5 | MC TNFRSF14 contributes to OVA-induced asthma pathology.** (**a**) Changes in $R_L$ induced by aerosolized methacholine, (**b**) numbers of leukocytes in BAL fluid, (**c,d**) levels of plasma OVA-specific IgG$_1$ (**c**) and IgE (**d**), (**e**) lung collagen and (**f**) lung MCs, 24 h after the ninth OVA (or PBS) challenge in *kit* mutant MC-deficient mice ($Kit^{W-sh/W-sh}$) engrafted with $Tnfrsf14^{+/+}$ or $Tnfrsf14^{-/-}$ BMCMCs ($Tnfrsf14^{+/+}$ BMCMCs→$Kit^{W-sh/W-sh}$ and $Tnfrsf14^{-/-}$ BMCMCs→$Kit^{W-sh/W-sh}$). Four-to-fifteen female mice per group were used as indicated. Results are pooled from three independent experiments, each of which gave similar results. (**g**) Lung sections stained with haematoxylin and eosin (H&E) (upper panel), Masson's Trichrome (middle panel), demonstrating collagen (blue staining) and goblet cells (small arrows) or Toluidine blue (lower panel), demonstrating MCs (arrows) 24 h after the ninth OVA challenge. Scale bar, 100 μm (insets: 25 μm). The data in **a–f** (shown as mean + or ± s.e.m.) were assessed for statistical significance using a two-tailed Student's *t*-test or a two-way analysis of variance (ANOVA) test. Asterisks indicate statistical significance of differences between PBS-treated and corresponding Ag-treated groups; daggers indicate statistical significance between indicated groups. * or †$P < 0.05$; ** or ††$P < 0.01$; *** or †††$P < 0.001$.

asthma[25,36] and a model induced by HDM. Our data indicate that the features of these asthma models that were significantly influenced by TNFRSF14 include blood levels of Ag-specific IgE, airway inflammation, AHR, lung collagen content and airway goblet cell hyperplasia and BAL Muc5AC content.

Because TNFRSF14 can influence T$_H$2 populations[24], we considered whether the effects of TNFRSF14 on the phenotypic

features of the asthma models in $Tnfrsf14^{-/-}$ mice could reflect, at least in part, actions of TNFRSF14 upstream of Ag-specific IgE production. However, we showed that the amounts of Ag-specific IgE produced by OVA-immunized $Tnfrsf14^{-/-}$ mice were sufficient to passively sensitize naive mice for the expression of airway responses to intra-tracheal Ag challenge. Moreover, we found that antagonizing TNFRSF14:TNFSF14 interactions using

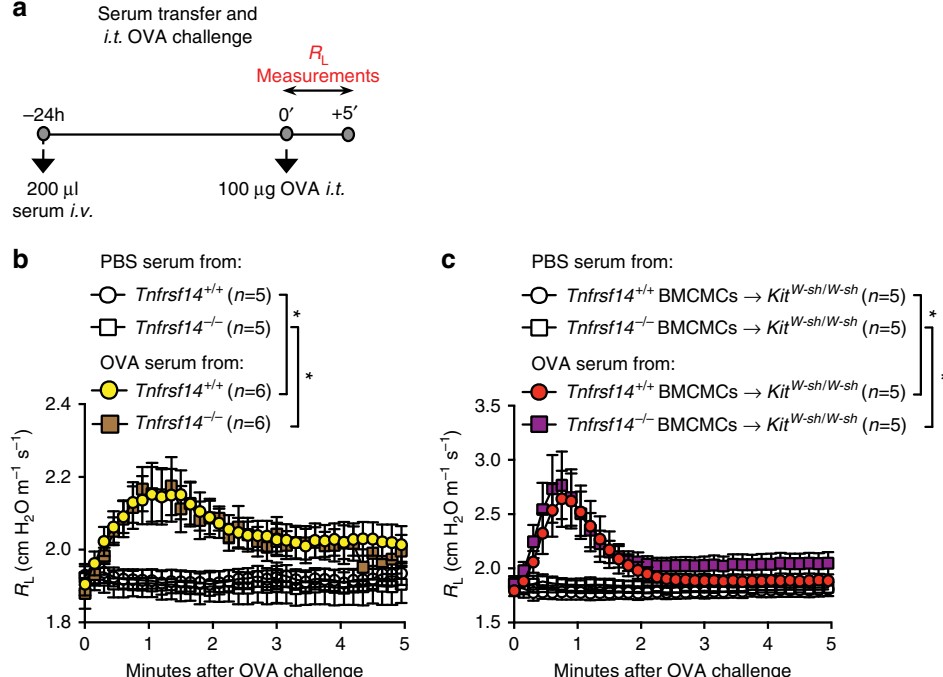

**Figure 6 | Passive transfer of Ag responsiveness with serum from Tnfrsf14−/− mice. (a)** Experimental protocol. Naive mice received i.v. 200 µl of serum from mice sensitized with OVA ('OVA serum') or mock-sensitized with PBS ('PBS serum') as shown in Fig. 2a, and then were challenged 24 h later with 100 µg OVA intra-tracheally (i.t.). **(b,c)** Lung resistance ($R_L$) was monitored for 5 min after Ag (OVA) stimulation in mice that received **(b)** serum from Tnfrsf14+/+ or Tnfrsf14−/− mice or **(c)** serum from $Kit^{W-sh/W-sh}$ mice engrafted i.v. with Tnfrsf14+/+ or Tnfrsf14−/− BMCMCs (in the latter mice, only MCs lacked TNFRSF14). Five-to-six female mice per group were used as indicated. Data are pooled from two independent experiments. The data in **b,c** (shown as mean ± s.e.m.) were assessed for statistical significance using a two-way analysis of variance (ANOVA) test. Asterisks indicate statistical significance of differences between indicated groups. *$P < 0.05$.

an TNFRSF14 blocking antibody administered to WT mice weeks after the period of Ag sensitization significantly reduced multiple features of the asthma pathology that develop in this setting, including AHR, lung inflammation and several measures of airway remodelling. Taken together, these findings strongly suggest that TNFRSF14:TNFSF14 interactions can influence the pathology of asthma models by mechanisms in addition to effects on the production of Ag-specific IgE.

Like other members of the TNF:TNFR superfamily, TNFRSF14 can function as co-stimulatory molecule[21], and TNFRSF14 is widely expressed in both lymphoid and myeloid cells[42]. We discovered that both mouse and human MCs express TNFRSF14 but not the alternative TNFSF14 receptor, TNFSF3. Moreover, despite the broad cellular distribution of TNFRSF14, our experiments employing two types of MC-deficient mice which had been engrafted with MCs that did or did not express TNFRSF14 showed that TNFRSF14 expression specifically on MCs is critical for the full development of multiple features of asthma pathology in vivo. Taken together with our in vitro findings indicating that TNFSF14 co-stimulation of MCs via TNFRSF14 can markedly increase both the numbers of aggregates of anti-IgE:FcεRI or specific-antigen:FcεRI on the surface of MCs stimulated with anti-IgE or specific antigen, respectively, as well as the IgE-dependent activation of the Akt, PKC-δ and MAPK cascade in MCs, which is accompanied by enhanced MC production of pre-formed, lipid and newly synthesized mediators, these findings indicate that TNFRSF14:TNFSF14 interactions may have a major role in fine tuning the extent of MC activation by IgE- and Ag in the asthma models we studied and perhaps in other settings in vivo.

For example, FcεRI/TNFRSF14 co-activation of MCs enhanced activation of Akt, PKC-δ, and the MAPK cascade, which is associated with the increased production of pro-inflammatory mediators (for example, TNF-α and IL-6) and pro-remodelling compounds (for example, leukotrienes and IL-13) (refs 33–35). Such MC-derived mediators not only may directly promote inflammation and tissue remodelling, but MC secretion of IL-13 also could enhance ongoing $T_H2$ responses and IgE production[8,31,43]. Moreover, some of the effects of MC activation in this setting may reflect actions of MC-derived mediators on other cell types. For example, Doherty et al.[20] proposed that increased production of both IL-13 by eosinophils (via TNFSF14 stimulation of eosinophil TNFRSF14) and TGF-β by macrophages (via TNFSF14 stimulation of macrophage TNFRSF3) might represent important TNFSF14-mediated mechanisms which contribute to airway remodelling. We think it is likely that MCs can influence the recruitment, activation and function of eosinophils and macrophages, as well as many other effector and target cells, in this setting[31]. Specifically, various MC mediators can activate eosinophils (for example, TNF-α and leukotrienes[44]), macrophages (for example, IL-13 (ref. 45)), neutrophils (for example, TNF-α[46]), ILC2 cells (for example, leukotrienes[47]) and lung structural cells (for example, histamine, leukotrienes and TNF-α[48,49]). Indeed, by being one of the first cells to respond to Ag challenge (via IgE bound to their FcεRI) and by also being responsive to TNFSF14, which might be produced at sites of developing inflammation by many cells, including various myeloid or lymphoid cells[20,50,51] that can be recruited in part in response to MC-derived mediators, MCs might function as powerful local amplifiers of airway responses to Ag.

It is not clear to what extent the IgE-dependent activation of MCs in lung tissues is regulated by soluble versus membrane-associated forms of TNFSF14. The ELISA we used to quantify TNFSF14 in BAL fluid only measures soluble TNFSF14. The data

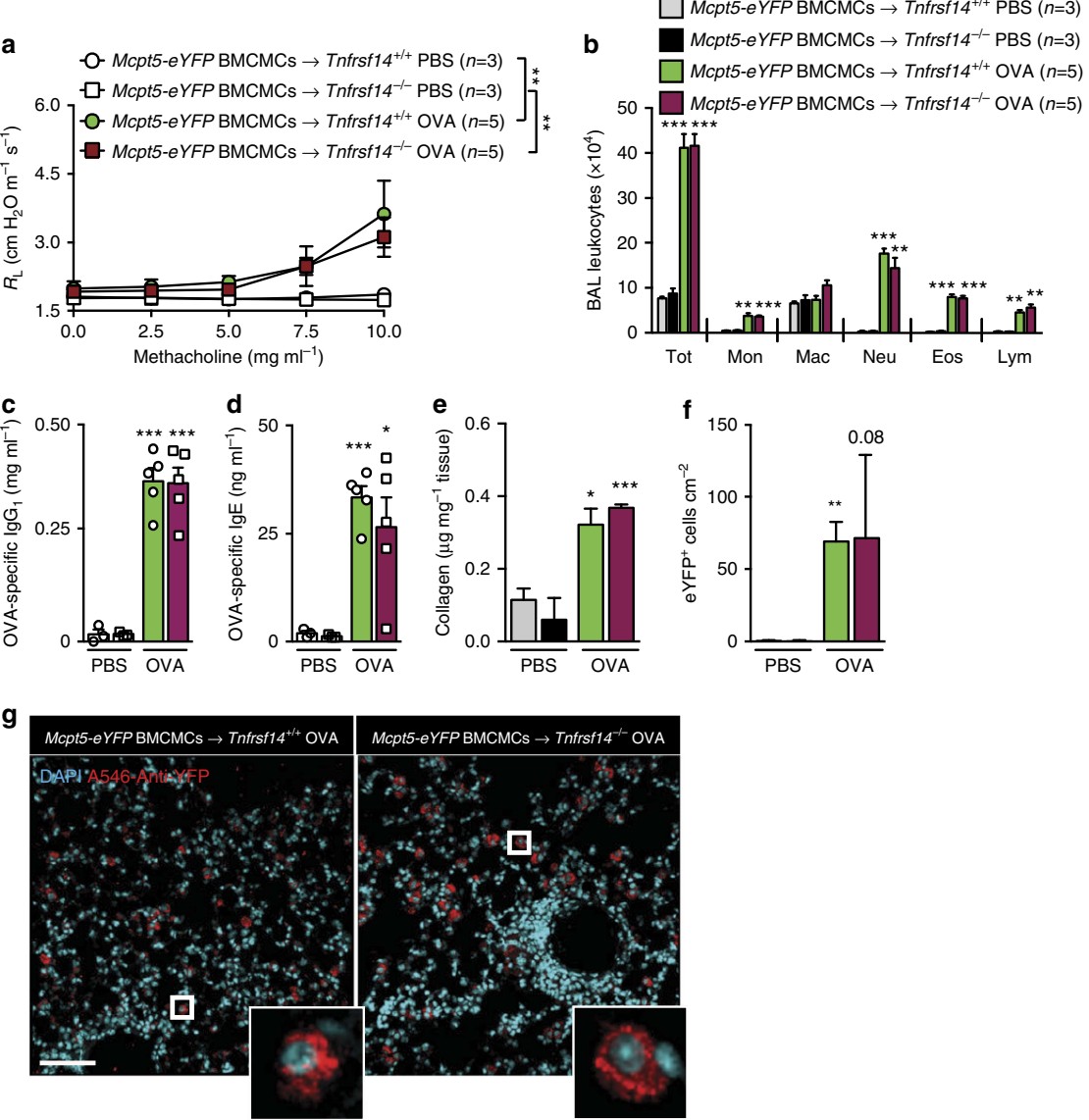

**Figure 7 | MC TNFRSF14 is sufficient for OVA-induced asthma pathology.** (**a**) Changes in $R_L$ induced by aerosolized methacholine, (**b**) numbers of leukocytes in BAL fluid, (**c,d**) levels of plasma OVA-specific IgG$_1$ (**c**) and IgE (**d**), (**e**) lung collagen and (**f**) numbers of engrafted eYFP$^+$ mast cells in the lung, 24 h after the ninth OVA (or PBS) challenge in $Tnfrsf14^{+/+}$ or $Tnfrsf14^{-/-}$ mice engrafted with Mcpt5-eYFP BMCMCs (Mcpt5-eYFP BMCMCs→$Tnfrsf14^{+/+}$ and Mcpt5-eYFP BMCMCs→$Tnfrsf14^{-/-}$). Three-to-five female mice per group were used as indicated. Results are pooled from two independent experiments, each of which gave similar results. (**g**) Lung sections stained with 4,6-diamidino-2-phenylindole and anti-YFP Ab, demonstrating engrafted lung MCs (eYFP$^+$ cells, indicated by red colour) in $Tnfrsf14^{+/+}$ or $Tnfrsf14^{-/-}$ mice engrafted with Mcpt5-eYFP BMCMCs (insets show images of representative eYFP$^+$ MCs). Scale bar, 50 µm. The data in **a–f** (shown as mean + or ± s.e.m.) were assessed for statistical significance using a two-tailed Student's $t$-test or a two-way analysis of variance (ANOVA) test. Asterisks indicate statistical significance of differences between PBS-treated and corresponding Ag-treated groups. *$P < 0.05$; **$P < 0.01$; ***$P < 0.001$.

from our analyses of TNFSF14 in BAL fluids, which are shown in Supplementary Figs 5e, 6e, 7e, 8j, 9c and 10c,f, indicate that the concentration of TNFSF14 we used *in vitro* to stimulate mouse BMCMCs and huPBMCs was substantially greater than that we measured in BAL fluids *in vivo*.

A number of factors may have contributed to this discrepancy. It has been reported that recombinant TNFSF14 is not stable and tends to aggregate, which potentially can influence the results of *in vitro* experiments[52]. *In vivo*, the biologically important TNFSF14 in airway inflammation is that which is present in immediate proximity to its receptors. Unfortunately, it is not possible directly to measure interstitial amounts of soluble TNFSF14. Moreover, TNFSF14 can bind to its receptor in

either its membrane-associated or soluble form[20]. It is not possible to quantify directly the amount of membrane-associated TNFSF14 in tissues and, despite much effort, we have not found an anti-TNFSF14 antibody that can be reliably used for IHC. Based on these considerations, we think that the amounts of TNFSF14 measured in the BAL fluids, which is soluble TNFSF14 that can be washed out of the lungs, may not reflect the physiologically or pathologically relevant amounts of TNFSF14 found in proximity to MCs in the tissues. Even the amounts of soluble TNFSF14 measurable in BAL fluid by ELISA appear to be quite variable—as one can see by examining the data from the various groups of OVA sensitized and challenged 'wild-type' mice tested in our different experiments. While there might be several

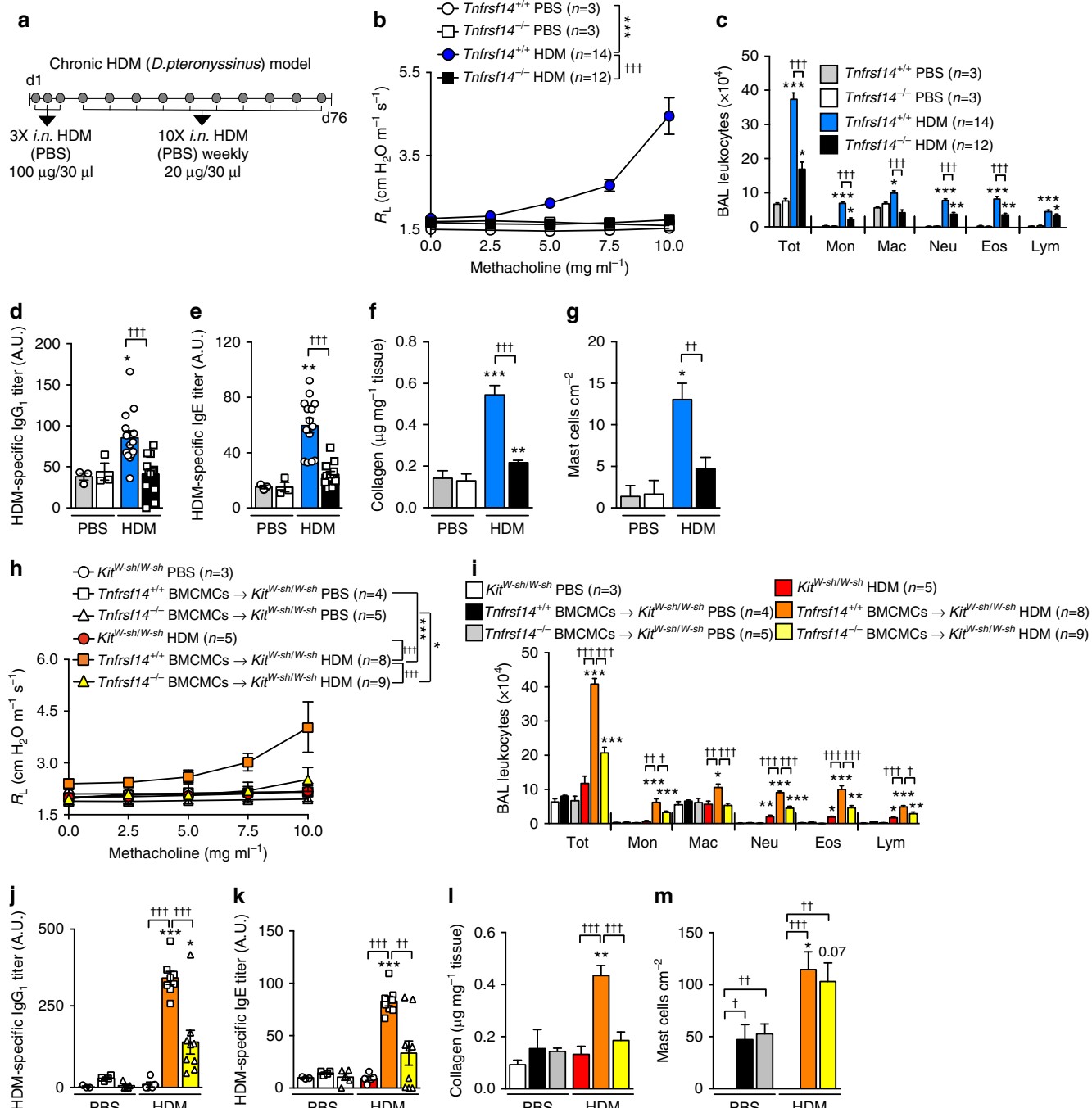

**Figure 8 | MC TNFRSF14 contributes to HDM-induced asthma pathology.** (**a**) Protocol for inducing HDM (*D. pteronyssinus*) sensitization and HDM-induced airway pathology. *Tnfrsf14*$^{-/-}$ and littermate control *Tnfrsf14*$^{+/+}$ mice were sensitized with three intranasal (i.n.) administrations of HDM followed by 10 weekly i.n. challenges with HDM; control mice were mock-sensitized with PBS, followed by 10 weekly intranasal (i.n.) challenges with PBS. (**b**) Changes in $R_L$ induced by aerosolized methacholine, (**c**) numbers of leukocytes in BAL fluid, (**d,e**) titers of plasma HDM-specific $IgG_1$ (**d**) and IgE (**e**), (**f**) lung collagen and (**g**) numbers of lung MCs, 24 h after the tenth HDM or PBS challenge in *Tnfrsf14*$^{-/-}$ and *Tnfrsf14*$^{+/+}$ mice. (**h–m**) Changes in $R_L$ (**h**) induced by aerosolized methacholine, (**i**) numbers of leukocytes in BAL fluid, (**j,k**), titers of plasma HDM-specific $IgG_1$ (**j**) and IgE (**k**), (**l**) lung collagen and (**m**) numbers of lung MCs, 24 h after the tenth HDM (or PBS) challenge in MC-deficient *Kit*$^{W-sh/W-sh}$ mice and in *Kit*$^{W-sh/W-sh}$ mice engrafted with *Tnfrsf14*$^{+/+}$ or *Tnfrsf14*$^{-/-}$ BMCMCs. Three-to-fourteen female mice per group were used as indicated. Results are pooled from at least two independent experiments, each of which gave similar results. The data in **b–m** (shown as mean + or ± s.e.m.) were assessed for statistical significance using a two-tailed Student's *t*-test or a two-way analysis of variance (ANOVA) test. Asterisks indicate statistical significance of differences between PBS-treated and corresponding Ag-treated groups; daggers indicate statistical significance between indicated groups. * or $^{\dagger}P < 0.05$; ** or $^{\dagger\dagger}P < 0.01$; *** or $^{\dagger\dagger\dagger}P < 0.001$.

reasons for this variability, the finding makes us wonder how reliable such measurements are in reflecting differences in the levels of the protein (in either the soluble or membrane-associated form) that engage in potentially important interactions

with tissue MCs. Indeed, one of the reasons we elected to use a genetic approach to investigate the importance of TNFRSF14, and MC expression of TNFRSF14, in features of asthma pathology is because of the difficulty in inferring biological importance for

specific mediators, receptors or cells solely on the basis of *in vitro* studies, IHC and so on. By studying the phenotype of our asthma models in mice that did or did not express TNFRSF14, and that did or did not contain MCs that expressed TNFRSF14, we avoided problems associated with drawing conclusions from *in vitro*, histological or IHC analyses taken in isolation.

In summary, we have identified TNFRSF14 as an important new co-receptor of mouse and human MCs, and showed that engagement of MC TNFRSF14 by TNFSF14 can significantly enhance IgE-dependent MC activation and mediator production. TNFRSF14 expression by MCs also can exacerbate several features of the pathology in two different mouse models of Ag-induced asthma *in vivo*. Moreover, in WT mice, blocking TNFRSF14:TNFSF14 interactions with an antibody administered weeks after Ag sensitization can diminish multiple features of asthma pathology, including blood levels of Ag-specific IgE, airway inflammation and AHR, and airway remodelling. Taken together with correlative clinical observations showing that levels of TNFSF14 in the sputum of a large group of subjects with asthma were inversely correlated with a measure of lung function[19], our findings support the notion that specific blockade of TNFRSF14, or combined targeting of TNFRSF14 and MC activation via the FcεRI, might have benefit in the treatment of some subjects with allergic asthma.

## Methods

**Mice.** All animal experiments were conducted according to the National Institutes of Health (NIH) guidelines and with the approval of the Stanford University Institutional Animal Care and Use Committee. Six to 8-week-old female C57BL/6J mice were used in experiments involving solely WT mice. Transgenic mouse strains were bred and housed with the respective control mice in the local animal facilities. $Tnfrsf14^{-/-}$ mice from Mitchell Kronenberg were backcrossed onto C57BL/6J mice at Stanford for 8–9 generations. Three-to-fifteen female naive mice (of the appropriate genotype and age) that were used in individual experiments were assigned randomly to the experimental groups (for example, those treated with OVA or HDM, or with anti-TNFRSF14 antibody, isotype control antibody or PBS).

**Antibodies.** All antibodies were used at $1 \mu g\,ml^{-1}$ when the manufacturer provided information about the initial concentration of the preparation of antibody purchased. For other antibodies for which this information was not provided by the manufacturer, the antibody preparation was diluted before use according to the manufacturer's instructions.

*Human.* Fluorophore conjugated anti-TNFRSF3 antibody (31G4D8) was from BioLegend and anti-TNFSF14 (7-3 7), anti-TNFRSF14 (eBioHVEM-122) and LAMP-1 (eBioH4A3) antibodies were from eBioscience.

*Mouse.* Anti-TNFRSF14 and fluorophore-conjugated anti-TNFRSF14 (both LH1), anti-TNFRSF3 (eBio3C8), anti-CD117 (2B8), anti-CD4 (GK1.5), anti-interferon (IFN)-γ (XMG1.2), anti-IL-4 (11B11), anti-IL-17 (eBio17B7) and anti-LAMP-1 (1D4B) antibodies were purchased from eBioscience. For staining of mTNFSF14, we blocked LTβ with an anti-LTβ antibody (FL-244, Santa Cruz Biotech), followed by incubation with TNFRSF3-Fc (from R&D) and fluorophore-conjugated anti-mouse IgG2a antibody (eBioscience). The same results (that is, no detection of TNFSF14 on the surface of the cells) were obtained when we performed flow cytometry using a fluorophore-conjugated rabbit polyclonal anti-mouse TNFSF14 antibody from Bioss (cat. no. bs-2462R-PE). Fluorophore-conjugated anti-FcεRI (MAR-1) was from Biolegend. Polyclonal anti-phospho-PKC-δ (T311), anti-phospho-Akt (T308, clone D25E6), anti-phospho-Mek1/2 (S217/S221, 41G9), polyclonal anti-phospho Erk1/2 (T202/Y204), anti-PKC-δ (D10E2), anti-Akt (C67E7), anti-Mek1/2 (47E6) and polyclonal anti-Erk1/2 antibodies were all from Cell Signaling Technology.

**Western blotting.** In all, $10 \times 10^6$ IgE pre-sensitized BMCMCs for each condition tested were activated as described above. At indicated time points, BMCMCs were lysed and total protein extracts were resolved onto a polyacrylamide gel. Blotted extracts were probed with indicated antibodies. Detection of Ras was performed with $20 \times 10^6$ IgE pre-sensitized BMCMC for each condition, using the Active Ras Pull-down kit (Thermo Scientific), according to the manufacturer's instructions.

**LAD2 human MC line and generation of cultured human MCs.** The LAD2 human MC line was kindly provided by Dr A. Kirshenbaum (NIH, USA) and cultured as previously described[53]. huPBCMCs were derived *in vitro* from peripheral blood $CD34^+$ mononuclear cells as previously reported[54].

**Engraftment of BMCMCs.** BMCMCs were obtained through *in vitro* differentiation of bone marrow precursors in WEHI-3-conditioned DMEM, for at least 5 weeks[55]. Bone marrow cells derived from 4-week-old $Tnfrsf14^{-/-}$, $Tnfrsf14^{+/+}$ or *Mcpt5-eYFP* mice were cultured in WEHI-3-conditioned DMEM (DMEM containing 20% supernatant of WEHI-3 cells, 10% FBS, $50 \mu M$ β-mercaptoethanol, 2 mM L-glutamine and 1% antibiotic-antimycotic solution), as a source of IL-3, for 4–5 weeks to generate cell populations that contained >99% bone-marrow-derived cultured MCs (BMCMCs). BMCMCs ($2 \times 10^6$) were injected into each mouse via the tail vein, and the recipients (for example, $Tnfrsf14^{-/-}$ BMCMCs→$Kit^{W-sh/W-sh}$, $Tnfrsf14^{+/+}$ BMCMCs→$Kit^{W-sh/W-sh}$, $Tnfrsf14^{-/-}$ BMCMCs→$Cpa3$-$Cre;Mcl$-$1^{fl/fl}$, $Tnfrsf14^{+/+}$ BMCMCs→$Cpa3$-$Cre;Mcl$-$1^{fl/fl}$ mice, *Mcpt5-eYFP* BMCMCs→$Tnfrsf14^{-/-}$ and *Mcpt5-eYFP* BMCMCs→$Tnfrsf14^{+/+}$) were used for experiments 8 weeks later.

**Activation of BMCMCs and huPBCMCs.** huPBCMCs were pre-sensitized with $1 \mu g\,ml^{-1}$ human IgE (Millipore) overnight, washed and stimulated with $2 \mu g\,ml^{-1}$ rabbit anti-human IgE (Bethyl). For IgE pre-sensitization and activation of BMCMCs, dinitrophenyl (DNP)-specific IgE (clone ε26 (ref. 56) was provided by Dr Fu-Tong Liu (University of California-Davis) and p-nitrophenyl-N-acetyl-β-D-glucosaminide, dinitrophenyl-conjugated human serum albumin (DNP-HSA) was obtained from Sigma. BMCMCs were pre-sensitized overnight with $1 \mu g\,ml^{-1}$ IgE in DMEM 5% FBS and stimulated with DNP-HSA (Ag) at a final concentration of $10\,ng\,ml^{-1}$.

**ELISA measurements from supernatants.** IgE pre-sensitized huPBCMCs were stimulated with anti-IgE as described above for the time indicated in the main text. In some cases huPBCMCs received $10 \mu g\,ml^{-1}$ recombinant human TNFSF14 (R&D Systems) at the same time as anti-IgE or PBS. Human IL-8 and TNF-α ELISA kits were from eBioscience.

IgE pre-sensitized BMCMCs were stimulated with Ag as described above for the time indicated in the main text. In some cases BMCMCs received $10 \mu g\,ml^{-1}$ recombinant mouse TNFSF14 (R&D Systems) at the same time as Ag or PBS. Histamine in the BMCMC culture supernatants was measured using an Enzyme Immunoassay Kit (Beckman Coulter) according to the manufacturer's instructions. ELISA kits for TNF-α, IL-6 and IL-13 were purchased from eBioscience. ELISA kits for $LTC_4$ and $LTE_4$ were purchased from Cayman Chemicals.

**Cytofluorimetric detection of LAMP-1 on surface of MCs.** IgE pre-sensitized BMCMCs or huPBCMCs were stimulated with PBS, Ag or anti-IgE (with or without TNFSF14), for 30 min. Cells were then incubated in ice for 5 min to block the reaction and stained for LAMP-1. Mean fluorescence intensity was measured for each BMCMC or huPBCMC and condition tested.

**Generation of Mcpt5-eYFP BMCMCs.** *Mcpt5-Cre⁺* mice were provided by Axel Roers[57] and were crossed to Cre-inducible ROSA-eYFP reporter mice (B6.129X1-$Gt(ROSA)26Sor^{tm1(EYFP)Cos}$/J; Jackson) to generate *Mcpt5-eYFP* BMCMCs.

**OVA asthma model and anti-TNFRSF14 antibody treatment.** All experiments employed OVA preparations that contained low levels of LPS ([$<0.1\,EU\,ml^{-1}$, measured with LAL Endotoxin Assay kit (Genscript)]).

Mice were immunized by three intraperitoneal injections of $50 \mu g$ OVA (Sigma-Aldrich) in $100 \mu l$ PBS on days 1, 4 and 7 (ref. 25). Starting on day 12, mice were challenged i.n. with $20 \mu g$ OVA in $30 \mu l$ PBS weekly for 9 weeks; control mice received i.n. challenges with PBS on the same schedule[25]. In some experiments, C57BL/6J mice received $20 \mu g$ of a hamster anti-TNFRSF14 antibody (clone LH1, eBioscience), an Armenian hamster IgG isotype control Ab (clone eBio299Arm, eBioscience) or PBS, 1 h before the eighth and the ninth i.n. OVA or PBS challenge.

**Immunization and airway challenge with HDM.** Mice were immunized by three i.n. challenges of $100 \mu g$ HDM from *D. pteronyssinus* (Greer) in $30 \mu l$ PBS on days 1, 4 and 7. Starting on day 12, mice were challenged i.n. with $20 \mu g$ HDM in $30 \mu l$ PBS weekly for 10 weeks; control mice received i.n. challenges with PBS on the same schedule.

**BAL fluid, cells and histology.** Twenty-four hours after the last OVA, HDM or PBS challenge mice were killed by $CO_2$ inhalation, the lungs were ligated, removed and lavaged with ice-cold Hanks' balanced salt solution (HBSS) (BAL). Cells were harvested, stained with Hema 3 Stain Set (Fisher Diagnostic) and counted on the basis of their morphological and staining properties. After recovery of BAL fluid, lungs were fixed (10% formaldehyde) and embedded in paraffin. Sections of $5 \mu m$ were mounted on Superfrost Plus glass slides (Fisher Scientific) and stained with haematoxylin and eosin, Masson's trichrome stains or toluidine blue.

**ELISA measurements from plasma and BAL fluid.** Plasma and BAL fluid were collected 24 h after the ninth OVA or PBS challenge or after the tenth HDM or PBS

challenge. ELISA kits for OVA-specific IgG$_1$ and IgE were purchased from Cayman Chemicals.

**HDM-specific IgG$_1$ and IgE titers.** Each incubation step in the ELISAs described below is followed by 3–5 washing steps using PBS containing 0.05% Tween-20. For detection of HDM-specific serum antibodies, MaxiSorp ELISAs plates (Nunc) were coated with 5 µg ml$^{-1}$ HDM at 4 °C overnight, followed by blocking with 1% bovine serum albumin in PBS for at least 2 h at room temperature. Sera diluted in PBS containing 1% bovine serum albumin were added and incubated in the blocked wells for 2 h at 37 °C. We detected bound IgG$_1$ and IgE antibodies using biotinylated detection antibodies (rat anti-mouse IgG$_1$ (clone A85-1, BD Pharmingen; incubated for 1 h at room temperature) and rat anti-mouse IgE (clone R35-118, BD Pharmingen), respectively), followed by incubation with horseradish peroxidase-conjugated streptavidin (BD Pharmingen) for 30 min at room temperature and detection using supersensitive 3,3',5,5'-tetramethylbenzidine (TMB) substrate (Sigma). Antibody titers were calculated by plotting the serum dilution that gave half-maximal signal of a reference serum. Measurements of mediators from BAL fluid was performed as described above.

**Measurement of lung collagen.** Lung collagen was extracted using acid pepsin method and levels of collagen were measured using a Sircol Soluble Collagen Assay Kit (Biocolor Life Science Assays).

**Passive immunization and measuring Ag responses in the lung.** Two-hundred millilitres of serum collected from $Tnfrsf14^{+/+}$ or $Tnfrsf14^{-/-}$ mice or $Kit^{W-sh/W-sh}$ mice that had been engrafted with $Tnfrsf14^{+/+}$ or $Tnfrsf14^{-/-}$ BMCMCs were injected i.v. per naive C57Bl/6 mouse. 24 h later, mice were anaesthetized, tracheotomized and challenged intra-tracheally with 100 µg OVA. $R_L$ was recorded over 5 min time.

**Time-lapse analysis of MC activation.** In all, $5 \times 10^4$ IgE pre-sensitized huPBCMCs or BMCMCs from $Tnfrsf14^{+/+}$ or $Tnfrsf14^{-/-}$ mice were placed into poly-D-Lysine coated micro-chambers (Nunc, 8 wells Lab-Tek) in 250 µl Tyrode's buffer supplemented with 1 µg ml$^{-1}$ of anti-human or mouse LAMP-1 (CD107a, a late endosomal/lysosomal marker used to monitor the dynamics of MC degranulation) antibody coupled to Alexa 488 (LAMP-1-A488). In some experiments, 10 µg ml$^{-1}$ human or mouse TNFSF14 coupled to Alexa 594 (TNFSF14-A594), 2 µg ml$^{-1}$ anti human-IgE (anti-IgE A650) (for experiments with human MCs) or 10 ng ml$^{-1}$ DNP-HSA coupled to Alexa 650 (DNP-HSA-A650) (for experiments with mouse MCs) and/or vehicle control were added to the medium. Fluorescence corresponding to LAMP-1-A488, TNFSF14-A594 and anti-IgE A650 or DNP-HSA-A650 was monitored simultaneously using a confocal laser-scanning microscope and in a controlled atmosphere (37 °C and 5% $CO_2$, Zeiss LSM780 Meta inverted, 63 × oil objective).

**3-D analysis of FcɛRI and TNFSF14 clusters.** BMCMCs or huPBCMCs were treated as described in the above Time-lapse analysis section. 30 min (for huPBCMCs) or 15 min (for BMCMCs) after stimulation with TNFSF14-A594, anti-IgE-A650 or DNP-HSA-A650, or both reagents, or vehicle alone (control), single-cell images were acquired as z stack (as optimal sections of 0.41 mm). 3-D reconstruction and modelling/analyses of clusters were performed using the Surface function of Imaris Bitplane Software. Because the software was unable to individually identify very large clusters that were in clone proximity on the plasma membrane, and, therefore, would identify them as a single large structure, we decided to omit modelled cluster area values for those few TNFSF14/anti-IgE clusters which reached the resolution limit that could be detected by the software (that is, areas $> 50 \mu m^2$) in depicting the areas of such clusters in Supplementary Fig. 1d,e. This approach, therefore, underestimated the upper limit of the modelled areas of TNFSF14/anti-IgE clusters in cells that contained very large clusters that were in close proximity on the plasma membrane.

**Reverse transcription–PCR.** Messenger RNA from BMCMCs and lungs was extracted using Trizol (Sigma). PCR was performed on complementary DNA prepared using the High Capacity Reverse Transcription Kit from Applied Biosystems, according to the manufacturer's instructions. Primer sequences for mouse TNFSF14 were: Fwd: ATAGTAGCTCATCTGCCAGATGGA; Rev: CGACTGACCAGCAGTTCTAACT.

**Measurements of airway reactivity.** Invasive measurements of airway reactivity in anaesthetized, tracheostomized, mechanically ventilated mice were performed by administering aerosolized Mch at increasing concentrations (0, 2.5, 5, 7.5, 10 mg ml$^{-1}$), with individual doses for 3 min, after a 2-min acclimation period, for a total for 5 min for each dose. $R_L$ and $C_{dyn}$ were continuously computed by fitting flow, volume and pressure to an equation of motion for each aerosol challenge period. Reported results were interpolated calculating the average $R_L$ and $C_{dyn}$ over a single time-period.

**T$_H$ profiling of cells in BAL fluid.** Mice were killed by $CO_2$ inhalation 24 h after the last OVA or PBS challenge and BAL was performed, and the cells in the BAL fluid were stimulated for 4 h with 50 ng ml$^{-1}$ phorbol 12-myristate 13-acetate (PMA), 500 ng ml$^{-1}$ ionomycin, and 2 µM monensin. Cells were then stained with an anti-CD4 antibody, fixed, permeabilized and stained for IFN-γ (T$_H$ 1), IL-4 (T$_H$ 2) and IL-17 (T$_H$17). Gated CD4$^+$ cells were analysed and quantified for their positivity for IFN-γ, IL-4 and/or IL-17 using a BD FACSCalibur cytofluorimeter. Data were analysed using FlowJo Data Analysis Software.

**ELISA measurements from plasma and BAL fluid.** ELISA kits for Mucin 5AC and TNFSF14 were purchased from Uscn Life Science Inc.

**Interpretation of results and statistics.** Evaluation of histology was performed in a blinded way, in that the observer was not aware of from which group of mice the slides were derived. For all the experiments, differences in airway responses between groups were tested for statistical significance using the two-way analysis of variance test. A Mann–Whitney test (for non-normal distribution) was used for Supplementary Fig. 1d,e and Supplementary Fig. 4d,e (note: for the calculations of the data shown in Supplementary Fig. 1d,e, five values in the anti-IgE + TNFSF14 condition (fourth columns) were excluded from the analysis since such TNFSF14/anti-IgE clusters reached the resolution limit that could be detected by the software); the unpaired Student's t-test was used for all other analyses. The unpaired Student's t-test was used for all other analyses. A P value $< 0.05$ was considered statistically significant. Asterisks indicate statistical significance between PBS (or non-Ag) and corresponding Ag-treated groups and daggers indicate statistical significance between indicated groups. * or $^\dagger P < 0.05$; ** or $^{\dagger\dagger} P < 0.01$; *** or $^{\dagger\dagger\dagger} P < 0.001$. All data are presented as mean + or ± s.e.m.

**Data availability.** Data supporting the findings of this study are available in the main article, the Supplementary Information Files, and/or from the authors upon request.

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

## Acknowledgements

This work was supported by US National Institutes of Health (NIH) grants to S.J.G. (U19AI104209 and R01AR067145), M.K. (R01AI61516) and L.L.R. (K99AI110645), fellowships from the Lucile Packard Foundation for Children's Health to R.S. (UL1 RR025744) and J.D.H. (UL1 TR001085), the Fondation pour la Recherche Medicale (FRM) SPE20130326582 and Philippe foundation to N.G., a Schroedinger Fellowship of the Austrian Science Fund (FWF) J3399-B21 to P.M.S., an NIH postdoctoral fellowship (2T32AI007290-31) to O.W.Z., the Department of Pathology and the Sean N. Parker Center for Allergy and Asthma Research, Stanford University.

## Author contributions

R.S., N.G., M. K. DeG., L.L.R., J.D.H., P.M.S., O.W.Z., S.B.M. and M.Y. helped to design (with S.J.G.) and/or perform the experiments. All authors helped to interpret data and edit the manuscript.

## Additional information

**Competing financial interests:** The authors declare no competing financial interests.

