## [Peer Review File · Nature Communications]

Reviewers' comments:

Reviewer #1 (Remarks to the Author):

In this manuscript authors have examined the role of an HVEM-FcERI mast cell pathway in the development of multiple features of asthma pathology in mice. The TNF family member LIGHT by interacting with its receptor, HVEM, supports TH2 cell generation and longevity, promotes airway remodeling in mouse models of asthma, although the mechanisms by which LIGHT functions in this setting are poorly understood. Here, authors show that blocking HVEM function reduces plasma levels of antigen specific IgG1 and IgE antibodies, airway hyper reactivity, airway inflammation and airway remodeling. This was done in mouse models by using a neutralizing antibody to HVEM or genetic deletion of HVEM. They also show using two types of mast cell deficient mice and reconstituted with mast cells that either express or lack the expression of HVEM on mast cells that HVEMs expression on mast cells significantly contributes to the development of multiple features of asthma biology. Overall the studies are well performed and conclusions are supported by the results. My major concern has to do with the lack of any data related to human mast cells beyond the first figure. Thus, while studies are well performed, the overall significance to human mast cells has not been looked at.

Reviewer #2 (Remarks to the Author):

Sibilano et al demonstrate that the LIGHT receptor HVEM in mast cells enhances antigen-induced mast cell responses and exacerbates several aspects of antigen-induced chronic airway inflammation. The paper is straight-forward, complete and well-written and presented. The authors used several in vivo models and transfer experiments to convincingly support their conclusions and the results are interesting and relevant. Overall, this is an excellent paper.

In fact, I do not see any major flaws with this paper, although I have a couple of comments/questions:

- 1- The concentrations of LIGHT used in the studies on mast cell responses seem high (10µg/ml). Have the authors tested what is the lowest concentration of LIGHT needed to enhance Ag-induced degranulation or any other mast cell response? Are the concentrations of LIGHT measurable in tissue and if so, are the levels in the range used in vitro?
- 2- A surprising observation is the effect of mast cell HVEM on OVA IgE and OVA IgG in the asthma model. In Figure 3, the data shows an effect of anti HVEM injection on OVA-IgG and E after 7 challenges with OVA. What are the kinetics of production of OVA specific Igs over successive challenges in this model?

Minor:

- 1- Typo in line 210: "FceRIg"
- 2- The order of figure 3 f,g,h and figure 4e,f,g do not correspond to the order it is cited in the text

Reviewer #3 (Remarks to the Author):

Summary: This is the first report that HVEM (herpes virus entry mediator) is expressed on human and mouse mast cells where it serves as a co-receptor for FcεRI when it is engaged by its ligand LIGHT (lymphotoxin-related inducible ligand). The authors demonstrate, by use of genetically altered mouse models (both knockouts and transgenic) and mast cells derived therefrom, that engagement of HVEM

potentiates IgE-mediated release of preformed and newly-formed mast cell mediators in vitro. Furthermore, HVEM expression substantially augments key pathological features of allergic asthma in different mouse models and, as demonstrated by adoptive transfer of mast cells to mast cell-deficient mice, mast-cell HVEM appears to be responsible in large part for the pathological effects. Given previous findings that asthma severity correlated with sputum levels of LIGHT in a large group of patients, the authors conclude that specifically targeting HVEM might be beneficial in the treatment of asthma.

Quality of work: The evidence for the conclusions drawn is substantial. All experimental findings are verified by alternative approaches and in total provide a sound basis for future therapeutic consideration. Statistics are adequate and appropriate. The manuscript is of sufficient quality and clarity for publication as is but the authors may wish to consider a few minor points.

- 1) The manuscript's focus is the mast cell but considering the broader readership of Nature Communications further clarification might be helpful for non-mast cell experts. For example, in addition to the examples of regulatory effects of TNF family members, of which LIGHT is a member (lines 66-72), it might be worth noting that ligands of the IL-1R super family including IL-33 and TLR ligands also have potent potentiating effects on IgE-mediated activation of mast cells to indicate the variety of regulatory factors influencing mast cell responses. The field has moved on since publication of the two reviews cited (10 and 11). Not necessary to state in this paper but it is interesting that both families rely on TRAF proteins for signaling and likely synergize via PI 3-kinase as the authors' data might indicate.
- 2) Likewise, an explanation that the Mcpt5 gene encodes for a mast cell-specific protease in granules would be helpful; hence the recognizable granules in the Figure 7H insets.
- 3) Line 165. Some readers may not know what is meant by "early TNFa"
- 4) Line 699, Figure 7 legend: should be "by" not "be".

RESPONSE TO THE REFEREES' COMMENTS

The authors thank the referees for their careful review of our manuscript and their constructive suggestions. Their advice has helped us to improve our manuscript. Here we provide a detailed point-by-point response to all of the referees' concerns.

Reviewers' comments:

Reviewer #1 (Remarks to the Author):

In this manuscript authors have examined the role of an HVEM-Fc ϵ RI mast cell pathway in the development of multiple features of asthma pathology in mice. The TNF family member LIGHT by interacting with its receptor, HVEM, supports TH2 cell generation and longevity, promotes airway remodeling in mouse models of asthma, although the mechanisms by which LIGHT functions in this setting are poorly understood. Here, authors show that blocking HVEM function reduces plasma levels of antigen specific IgG1 and IgE antibodies, airway hyper reactivity, airway inflammation and airway remodeling. This was done in mouse models by using a neutralizing antibody to HVEM or genetic deletion of HVEM. They also show using two types of mast cell deficient mice and reconstituted with mast cells that either express or lack the expression of HVEM on mast cells that HVEMs expression on mast cells significantly contributes to the development of multiple features of asthma biology. Overall the studies are well performed and conclusions are supported by the results. My major concern has to do with the lack of any data related to human mast cells beyond the first figure. Thus, while studies are well performed, the overall significance to human mast cells has not been looked at.

We wish to thank the reviewer for her/his comment "Overall the studies are well performed and conclusions are supported by the results". The only reservation expressed was that "the overall significance to human mast cells has not been looked at".

*While we are not able directly to study Fc ϵ RI:Ag:HVEM:LIGHT interactions on human mast cells (MCs) in vivo, we did perform additional studies of human MCs in vitro. We examined human peripheral blood cultured mast cells (huPBCMCs) and showed that LIGHT:HVEM interaction can synergize with Fc ϵ RI crosslinking, leading to the formation of Fc ϵ RI:Ag:HVEM:LIGHT clusters, resulting in enhanced MC activation responses. This mechanism appears to be similar to what we also have observed in mouse *Tnfrsf14*^{+/+} mast cells (but not *Tnfrsf14*^{-/-} mast cells, which lack HVEM and do not bind LIGHT).*

These findings are shown in new Supplementary Figures 1 and 3 and are described in the text in lines 140-170:

"In vitro stimulation with LIGHT in the absence of Fc ϵ RI-crosslinking did not detectably influence MC activation, suggesting that HVEM engagement can contribute to MC activation only in concert with another activation signal, in this case, Fc ϵ RI aggregation.

We also performed a single cell analysis of Fc ϵ RI and HVEM activation dynamics in living MCs in real time using time-lapse confocal laser scanning microscopy. We monitored, in 3-D and at high time resolution, granule secretion by huPBCMCs, as assessed by measuring the fluorescence of the granule-associated marker LAMP-130, using an AlexaFluor-conjugated anti-human (h)LAMP-1 Ab, LAMP-1-A488 (visualized in green), simultaneously with Fc ϵ RI and HVEM aggregation, using, respectively, AlexaFluor-conjugated anti-IgE [anti-IgE-A650] (visualized in blue) and AlexaFluor-conjugated LIGHT [LIGHT-A594] (visualized in red) (Supplementary Fig. 1a). When LIGHT-A594 was added to the huPBCMC cultures in the absence of Fc ϵ RI aggregation, only a modest number of HVEM/LIGHT-A594 aggregates were formed on the huPBCMC surface and, consistent with the data in Figure 1b-d, no LAMP-1 signals were detected (Supplementary Fig. 1a). The addition of anti-IgE-A650 induced formation of some Fc ϵ RI/anti-IgE-A650 clusters and rapid generation of LAMP-1 signals (Supplementary Fig. 1a). When anti-IgE and LIGHT were

added simultaneously, we observed formation of substantially higher numbers of both FcεRI/anti-IgE-A650 and HVEM/LIGHT-A594 clusters together with enhanced LAMP-1 fluorescence, indicating a synergistic effect of FcεRI-clustering and LIGHT on huPBCMC degranulation (Supplementary Fig. 1a).

To analyze precisely the number and dimension of clusters on the surface of individual huPBCMCs, we modeled both FcεRI/anti-IgE-A650 and HVEM/LIGHT-A594 clusters 30 min after stimulation. Combining anti-IgE-A650 with LIGHT dramatically increased both the number and area of individual clusters of FcεRI/anti-IgE-A650 and HVEM/LIGHT-A594 on the plasma membrane surface (Supplementary Fig. 1b-e)."

*We performed a similar experiment using BMCMCs as detailed in lines 212-221: "We also modeled both FcεRI/Ag (DNP-HSA-A650, blue) and HVEM/LIGHT-A594 (red) clusters after stimulation of mouse BMCMCs with LIGHT, IgE and antigen, or both stimuli. Exposure of IgE-sensitized mouse BMCMCs to both DNP-HSA-A650 and LIGHT dramatically increased both the number and area of individual clusters formed by FcεRI and HVEM on the plasma membrane surface (Supplementary Fig. 3a-e). Moreover, we did not observe any binding of LIGHT on the surface of *Tnfrsf14*^{-/-} BMCMCs, indicating that LIGHT binding is specific for MCs expressing HVEM (not shown). Taken together, our results with both human PBCMCs and mouse BMCMCs indicate that engagement of LIGHT by HVEM on the MC surface can enhance the IgE-dependent aggregation of FcεRI".*

We think that showing a similar mechanism for HVEM-mediated enhancement of IgE-dependent activation of human and mouse MCs in vitro, and providing a series of mechanistic experiments in two pre-clinical models of severe asthma, make a strong case for the hypothesis that HVEM, and in particular MC-expression of HVEM, might be relevant targets for therapeutic intervention in chronic asthma. We agree with the reviewer that more work will be needed to elucidate the extent of the involvement of MC-HVEM in the human pathology, particularly if a suitable therapeutic agent were to become available for clinical testing. However, we think that such work is beyond the scope of this manuscript.

Reviewer #2 (Remarks to the Author):

Sibilano et al demonstrate that the LIGHT receptor HVEM in mast cells enhances antigen-induced mast cell responses and exacerbates several aspects of antigen-induced chronic airway inflammation. The paper is straight-forward, complete and well-written and presented. The authors used several in vivo models and transfer experiments to convincingly support their conclusions and the results are interesting and relevant. Overall, this is an excellent paper.

In fact, I do not see any major flaws with this paper, although I have a couple of comments/questions:

1- The concentrations of LIGHT used in the studies on mast cell responses seem high (10µg/ml). Have the authors tested what is the lowest concentration of LIGHT needed to enhance Ag-induced degranulation or any other mast cell response? Are the concentrations of LIGHT measurable in tissue and if so, are the levels in the range used in vitro?

We thank the reviewer for her/his enthusiastic general comments about our manuscript, and for describing our work as "excellent".

To address the referee's first point, we now provide quantification of BAL LIGHT, as measured by ELISA (which only measures the LIGHT's soluble portion released in the BAL) (these data are shown in Supplementary Figures 5e, 6e, 7e, 8j, 9c, 10c, 10f).

These data indicate that the concentration of LIGHT we used in vitro to stimulate mouse BMCMCs and huPBMCs was substantially greater than that we measured in BAL fluids in vivo.

There might be a number of factors that can explain this discrepancy, as discussed in detail in the revised text of the Discussion (lines 472-506):

"It is not clear to what extent the IgE-dependent activation of MCs in lung tissues is regulated by soluble vs. membrane-associated forms of LIGHT. The ELISA we used to quantify LIGHT in BAL fluid only measures soluble LIGHT. The data from our analyses of LIGHT in BAL fluids, which are shown in Supplementary Figs. 5e, 6e, 7e, 8j, 9c, 10c, and 10f, indicate that the concentration of LIGHT we used in vitro to stimulate mouse BMCMCs and huPBMCs was substantially greater than that we measured in BAL fluids in vivo.

There may be a number of factors that can contribute to this discrepancy. It has been reported that recombinant LIGHT is not stable and tends to aggregate, which has the potential to influence the results of in vitro experiments⁵². In vivo, the biologically important LIGHT in airway inflammation is that which is present in immediate proximity to its receptors. Unfortunately, it is not possible directly to measure interstitial amounts of soluble LIGHT. Moreover, LIGHT can bind to its receptor in either its membrane-associated or soluble form²⁰. It isn't possible to quantify directly the amount of membrane-associated LIGHT in tissues and, despite much effort, we have not found an anti-LIGHT antibody that can be reliably used for IHC. Based on these considerations, we think that the amounts of LIGHT measured in the BAL fluids, which is soluble LIGHT that can be washed out of the lungs, may not reflect the physiologically or pathologically relevant amounts of LIGHT found in proximity to MCs in the tissues. Even the amounts of soluble LIGHT measurable in BAL fluid by ELISA appear to be quite variable – as one can see by examining the data from the various groups of OVA sensitized and challenged "wild type" mice tested in our different experiments. While there might be several reasons for this variability, the finding makes us wonder how reliable such measurements are in reflecting differences in the levels of the protein (in either the soluble or membrane-associated form) that engage in potentially important interactions with tissue MCs. Indeed, one of the reasons we elected to use a genetic approach to investigate the importance of HVEM, and MC expression of HVEM, in features of asthma pathology is because of the difficulty in inferring biological importance for specific mediators, receptors or cells solely on the basis of in vitro studies, IHC, etc. By studying the phenotype of our asthma models in mice that did or did not express HVEM, and that did or did not contain MCs that expressed HVEM, we avoided the problems associated with drawing conclusions from in vitro, histological or IHC analyses taken in isolation".

2- A surprising observation is the effect of mast cell HVEM on OVA IgE and OVA IgG in the asthma model. In Figure 3, the data shows an effect of anti HVEM injection on OVA-IgG and E after 7 challenges with OVA. What are the kinetics of production of OVA specific Igs over successive challenges in this model?

We performed an experiment in which we induced our OVA protocol in WT mice and collected blood 1 hour after each i.n. OVA challenge. At 8 weeks, the group of mice was divided into three groups, for no treatment, treatment with anti-HVEM antibody or treatment with the isotype control antibody, and blood was collected after the ninth i.n. OVA challenge. The results corroborate our initial findings and indicate that treatment with anti-HVEM antibody can reduce levels of OVA-specific IgG₁ and IgE antibodies even though the treatment occurs only during the last week of the protocol. The new data are presented in new Supplementary Figure 5f-i and are discussed in lines 280-291:

"These results were corroborated in a subsequent experiment in which blood was collected 1 h after each *i.n.* OVA challenge. After the seventh OVA challenge, the initial cohort of mice was split into 3 groups (no Ab treatment, anti-HVEM Ab treatment and Iso Ctrl Ab treatment) and at the end of the ninth *i.n.* OVA challenge, levels of OVA-specific IgG₁ and IgE collected during the nine *i.n.* challenges were quantified. Results shown in Supplementary Fig. 5f-i show that blockade of the HVEM:LIGHT axis during the last week of the model lowered the blood concentrations of OVA-specific IgG₁ and IgE Abs. These observations might reflect direct actions of the anti-HVEM Ab blockade on B cells²², as well as direct and/or indirect effect(s) of such treatment on other HVEM⁺ pro-inflammatory cell types, including MCs".

Minor:

1- Typo in line 210: "FceRIg"

We were referring to the gamma chain for the FcεRI (FcεRIγ). We have now clarified the abbreviation used (now in line 253-254).

2- The order of figure 3 f,g,h and figure 4e,f,g do not correspond to the order it is cited in the text

We have rearranged the text accordingly. Please see lines 259-268 and 297-301.

Reviewer #3 (Remarks to the Author):

Summary: This is the first report that HVEM (herpes virus entry mediator) is expressed on human and mouse mast cells where it serves as a co-receptor for FcεRI when it is engaged by its ligand LIGHT (lymphotoxin-related inducible ligand). The authors demonstrate, by use of genetically altered mouse models (both knockouts and transgenic) and mast cells derived therefrom, that engagement of HVEM potentiates IgE-mediated release of preformed and newly-formed mast cell mediators *in vitro*. Furthermore, HVEM expression substantially augments key pathological features of allergic asthma in different mouse models and, as demonstrated by adoptive transfer of mast cells to mast cell-deficient mice, mast-cell HVEM appears to be responsible in large part for the pathological effects. Given previous findings that asthma severity correlated with sputum levels of LIGHT in a large group of patients, the authors conclude that specifically targeting HVEM might be beneficial in the treatment of asthma.

Quality of work: The evidence for the conclusions drawn is substantial. All experimental findings are verified by alternative approaches and in total provide a sound basis for future therapeutic consideration. Statistics are adequate and appropriate. The manuscript is of sufficient quality and clarity for publication as is but the authors may wish to consider a few minor points.

1) The manuscript's focus is the mast cell but considering the broader readership of Nature Communications further clarification might be helpful for non-mast cell experts. For example, in addition to the examples of regulatory effects of TNF family members, of which LIGHT is a member (lines 66-72), it might be worth noting that ligands of the IL-1R super family including IL-33 and TLR ligands also have potent potentiating effects on IgE-mediated activation of mast cells to indicate the variety of regulatory factors influencing mast cell responses. The field has moved on since publication of the two reviews cited (10 and 11). Not necessary to state in this paper but it is interesting that both families rely on TRAF proteins for signaling and likely synergize via PI 3-kinase as the authors' data might indicate.

We thank the reviewer for her/his positive comments about our work. We have revised the Introduction so that we can mention examples of the regulatory effects of TNF members (lines 69-71), and by describing that several receptors, such as IL-33R, TSLPR and various TLRs, also can influence IgE-mediated activation of mast cells (lines 66-69).

We now have replaced original references 10 and 11 with more recent reviews (new refs 10 and 11).

2) Likewise, an explanation that the *Mcpt5* gene encodes for a mast cell-specific protease in granules would be helpful; hence the recognizable granules in the Figure 7H insets.

We have clarified this concern at lines 376-381 to read:

*"In *Mcpt5-Cre* transgenic mice, Cre is expressed under the control of the mast cell protease (*Mcpt*) 5 promoter. *Mcpt5-Cre* transgenic mice were crossed with Cre-inducible ROSA-eYFP*

reporter mice, as detailed in the Methods section. BMCMCs derived from such mice are thus HVEM+ (since they express a WT Tnfrsf14 gene) and specifically express eYFP".

3) Line 165. Some readers may not know what is meant by "early TNFa" .

We have explained the meaning of "early TNF- α " in line 198.

4) Line 699, Figure 7 legend: should be "by" not "be".

Thanks for noticing this error. We have now fixed this typo (now line 803).

REVIEWERS' COMMENTS:

Reviewer #1 (Remarks to the Author):

Authors have adequately addressed my concerns. The manuscript is significantly improved.

Reviewer #2 (Remarks to the Author):

Only a minor comment and a correction:

1- The answer to comment 1 of Reviewer #2 is valid. However, I feel it is a too lengthy explanation in the discussion and detracts from the main points. Lines 492 to 503 could be deleted without impacting the message of the paragraph.

2- In pg 9, line 186, shouldn't the reference to Figure 2a-h be Supplementary Figure 2a-h?

RESPONSE TO THE REFEREES' COMMENTS

The authors thank the referees for their supportive and constructive comments regarding our manuscript.

Reviewer #1 (Remarks to the Author):

Authors have adequately addressed my concerns. The manuscript is significantly improved.

Thank you for these comments.

Reviewer #2 (Remarks to the Author):

Only a minor comment and a correction:

1- The answer to comment 1 of Reviewer #2 is valid. However, I feel it is a too lengthy explanation in the discussion and detracts from the main points. Lines 492 to 503 could be deleted without impacting the message of the paragraph.

The authors think that this section is important and wish to keep in the main manuscript if the Editors agree. Also, we note that the other reviewer did not request that this section of the text be eliminated. The reasons we wish to retain the text are that: 1) we want readers to appreciate that we recognize that the LIGHT ELISA data are quite variable and therefore we have concerns about the utility of such measurements, and 2) we wish to comment on the power of using genetic and cell transfer approaches in such complex in vivo settings.

2- In pg 9, line 186, shouldn't the reference to Figure 2a-h be Supplementary Figure 2a-h?

The reviewer is correct. We have corrected our mistake.